# Can LLMs Facilitate Interpretation of Pre-trained Language Models?

**Basel Mousi**  **Nadir Durrani**  **Fahim Dalvi**

Qatar Computing Research Institute, HBKU, Doha, Qatar

{bmousi,ndurrani,faimaduddin}@hbku.edu.qa

## Abstract

Work done to uncover the knowledge encoded within pre-trained language models rely on annotated corpora or human-in-the-loop methods. However, these approaches are limited in terms of scalability and the scope of interpretation. We propose using a large language model, Chat-GPT, as an annotator to enable fine-grained interpretation analysis of pre-trained language models. We discover latent concepts within pre-trained language models by applying agglomerative hierarchical clustering over contextualized representations and then annotate these concepts using ChatGPT. Our findings demonstrate that ChatGPT produces accurate and semantically richer annotations compared to human-annotated concepts. Additionally, we showcase how GPT-based annotations empower interpretation analysis methodologies of which we demonstrate two: probing frameworks and neuron interpretation. To facilitate further exploration and experimentation in the field, we make available a substantial Concept-Net dataset (TCN) comprising 39,000 annotated concepts.[1]

## 1  Introduction

A large body of work done on interpreting pre-trained language models answers the question: *What knowledge is learned within these models?* Researchers have investigated the concepts encoded in pre-trained language models by probing them against various linguistic properties, such as morphological (Vylomova et al., 2017; Belinkov et al., 2017a), syntactic (Linzen et al., 2016; Conneau et al., 2018; Durrani et al., 2019), and semantic (Qian et al., 2016; Belinkov et al., 2017b) tasks, among others. Much of the methodology used in these analyses heavily rely on either having access to an annotated corpus that pertains to the linguistic concept of interest (Tenney et al., 2019; Liu et al.,

2019a; Belinkov et al., 2020), or involve human-in-the-loop (Karpathy et al., 2015; Kádár et al., 2017; Geva et al., 2021; Dalvi et al., 2022) to facilitate such an analysis. The use of pre-defined linguistic concepts restricts the scope of interpretation to only very general linguistic concepts, while human-in-the-loop methods are not scalable. *We circumvent this bottleneck by using a large language model, ChatGPT, as an annotator to enable fine-grained interpretation analysis.*

Generative Pre-trained Transformers (GPT) have been trained on an unprecedented amount of textual data, enabling them to develop a substantial understanding of natural language. As their capabilities continue to improve, researchers are finding creative ways to leverage their assistance for various applications, such as question-answering in financial and medical domains (Guo et al., 2023), simplifying medical reports (Jeblick et al., 2022), and detecting stance (Zhang et al., 2023). We carry out an investigation of whether GPT models, specifically ChatGPT, can aid in the interpretation of pre-trained language models (pLMs).

A fascinating characteristic of neural language models is that words sharing any linguistic relationship cluster together in high-dimensional spaces (Mikolov et al., 2013). Recent research (Michael et al., 2020; Fu and Lapata, 2022; Dalvi et al., 2022) has built upon this idea by exploring representation analysis through latent spaces in pre-trained models. Building on the work of Dalvi et al. (2022) we aim to identify encoded concepts within pre-trained models using agglomerative hierarchical clustering (Gowda and Krishna, 1978) on contextualized representations. The underlying hypothesis is that these clusters represent latent concepts, capturing the language knowledge acquired by the model. Unlike previous approaches that rely on predefined concepts (Michael et al., 2020; Durrani et al., 2022) or human annotation (Alam et al., 2023) to label these concepts, we leverage the ChatGPT model.

---

[1] https://neurox.qcri.org/projects/transformers-concept-net/

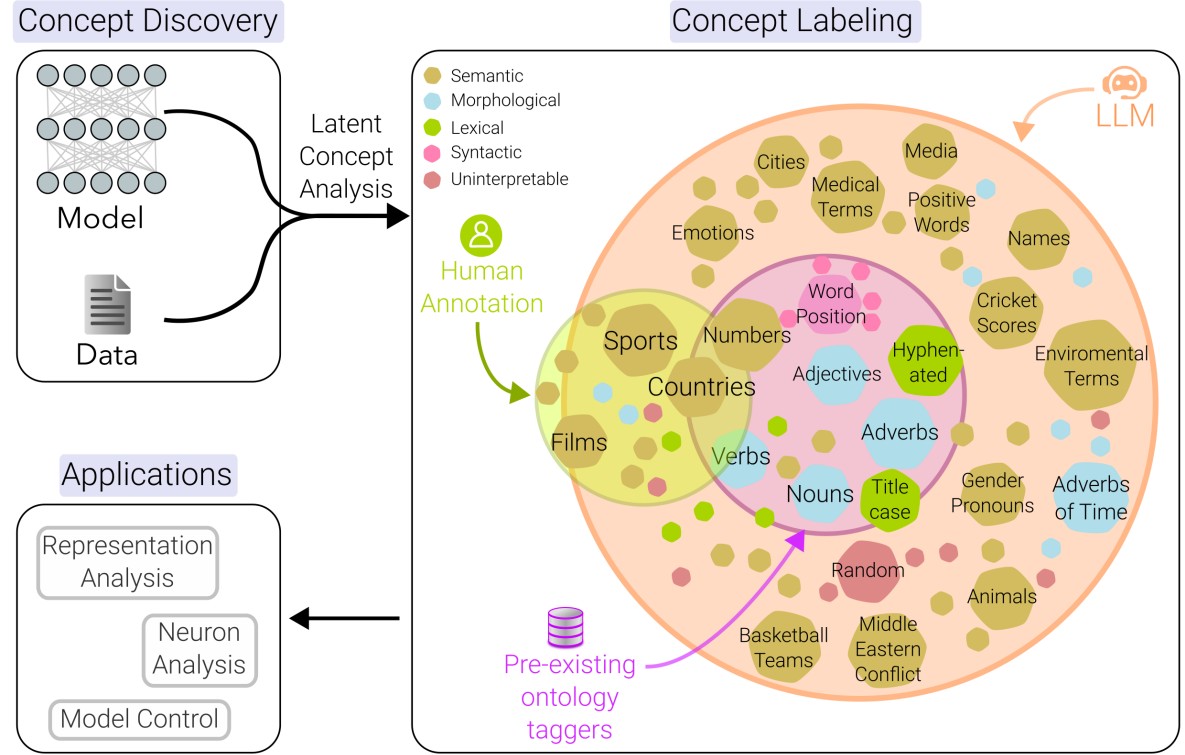

Figure 1: ChatGPT as an annotator: Human annotation or taggers trained on pre-defined concepts, cover only a fraction of a model's concept space. ChatGPT enables scaling up annotation to include nearly all concepts, including the concepts that may not have been manually annotated before.

Our findings indicate that the annotations produced by ChatGPT are semantically richer and accurate compared to the human-annotated concepts (for instance BERT Concept NET). Notably, Chat-GPT correctly labeled the majority of concepts deemed uninterpretable by human annotators. Using an LLM like ChatGPT improves scalability and accuracy. For instance, the work in Dalvi et al. (2022) was limited to 269 concepts in the final layer of the BERT-base-cased (Devlin et al., 2019) model, while human annotations in Geva et al. (2021) were confined to 100 keys per layer. Using ChatGPT, the exploration can be scaled to the entire latent space of the models and many more architectures. We used GPT to annotate 39K concepts across 5 pre-trained language models. Building upon this finding, we further demonstrate that GPT-based annotations empowers methodologies in interpretation analysis of which we show two: i) probing framework (Belinkov et al., 2017a), ii) neuron analysis (Antverg and Belinkov, 2022).

**Probing Framework** We train probes from GPT-annotated concept representations to explore concepts that go beyond conventional linguistic categories. For instance, instead of probing for named entities (e.g. NE:PER), we can investigate whether a model distinguishes between male and female names or probing for "Cities in the southeastern United States" instead of NE:LOC.

**Neuron Analysis** Another line of work that we illustrate to benefit from GPT-annotated latent concepts is the neuron analysis i.e. discovering neurons that capture a linguistic phenomenon. In contrast to the holistic view offered by representation analysis, neuron analysis highlights the *role of individual neurons* (or groups of them) within a neural network ((Sajjad et al., 2022). We obtain neuron rankings for GPT-annotated latent concepts using a neuron ranking method called Probeless (Antverg and Belinkov, 2022). Such fine-grained interpertation analyses of latent spaces enable us to see *how neurons distribute in hierarchical ontologies.* For instance, instead of simply identifying neurons associated with the POS:Adverbs, we can now uncover how neurons are distributed across sub-concepts such as adverbs of time (e.g., "tomorrow") and adverbs of frequency (e.g., "daily"). Or instead of discovering neurons for named entities (e.g. NE:PER), we can discover neurons that capture "Muslim Names" versus "Hindu Names".

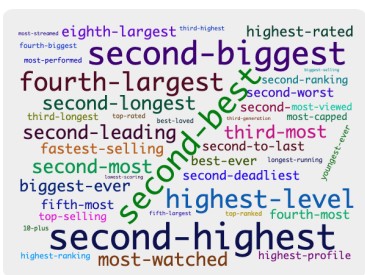
(a) Hyphenated Superlatives

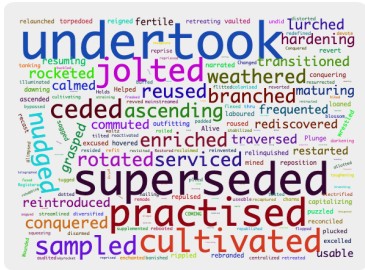
(b) Verbs

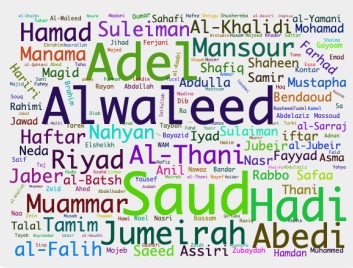
(c) Arab Names

Figure 2: Illustrative Examples of Concept Learned in BERT: word groups organized based on (a) Lexical, (b) Parts of Speech, and (c) Semantic property

To summarize, we make the following contributions in this work:

- Our demonstration reveals that ChatGPT offers comprehensive and precise labels for latent concepts acquired within pLMs.

- We showcased the GPT-based annotations of latent concepts empower methods in interpretation analysis by showing two applications: Probing Classifiers and Neuron Analysis.

- We release *Transformers Concept-Net*, an extensive dataset containing 39K annotated concepts to facilitate the interpretation of pLMs.

## 2 Methodology

We discover latent concepts by applying clustering on feature vectors (§2.1). They are then labeled using ChatGPT (§2.2) and used for fine-grained interpretation analysis (§2.3 and 2.4). A visual representation of this process is shown in Figure 1.

### 2.1 Concept Discovery

Contextualized word representations learned in pre-trained language models, can identify meaningful groupings based on various linguistic phenomenon. These groups represent concepts encoded within pLMs. Our investigation expands upon the work done in discovering latent ontologies in contextualized representations (Michael et al., 2020; Dalvi et al., 2022). At a high level, feature vectors (contextualized representations) are first generated by performing a forward pass on the model. These representations are then clustered to discover the encoded concepts. Consider a pre-trained model $\mathbf{M}$ with $L$ layers: $l_1, l_2, \ldots, l_L$. Using dataset $\mathbb{D} = w_1, w_2, \ldots, w_N$, we generate feature vectors $\mathbb{D} \xrightarrow{\mathbb{M}} \mathbf{z}^l = \mathbf{z}_1^l, \ldots, \mathbf{z}_n^l$.[2] Agglomerative hierar-

chical clustering is employed to cluster the words. Initially, each word forms its own cluster. Clusters are then merged iteratively based on Ward's minimum variance criterion, using intra-cluster variance as dissimilarity measure. The squared Euclidean distance evaluates the similarity between vector representations. The algorithm stops when $K$ clusters (encoded concepts) are formed, with $K$ being a hyper-parameter.

### 2.2 Concept Annotation

Encoded concepts capture latent relationships among words within a cluster, encompassing various forms of similarity such as lexical, syntactic, semantic, or specific patterns relevant to the task or data. Figure 2 provides illustrative examples of concepts encoded in the BERT-base-cased model.

This work leverages the recent advancements in prompt-based approaches, which are enabled by large language models such as GPT-3 (Brown et al., 2020). Specifically, we utilize a zero-shot learning strategy, where the model is solely provided with a natural language instruction that describes the task of labeling the concept. We used ChatGPT with zero-shot prompt to annotate the latent concepts with the following settings:[3]

```
Assistant is a large language model
trained by OpenAI
Instructions:
Give a short and concise label that best
describes the following list of words:
["word 1", "word 2", ..., "word N"]
```

### 2.3 Concept Probing

Our large scale annotations of the concepts in pLMs enable training probes towards fine-grained

---

[2]$z_i$ denotes the contextualized representation for word $w_i$

[3]We experimented with several prompts, see Appendix A.1 for details.

concepts that lack pre-defined annotations. For example we can use probing to assess whether a model has learned concepts that involve biases related to gender, race, or religion. By tracing the input sentences that correspond to an encoded concept $C$ in a pre-trained model, we create annotations for a particular concept. We perform fine-grained concept probing by extracting feature vectors from annotated data through a forward pass on the model of interest. Then, we train a binary classifier to predict the concept and use the probe accuracy as a qualitative measure of how well the model represents the concept. Formally, given a set of tokens $\mathbb{W} = \{w_1, w_2, ..., w_N\} \in C$, we generate feature vectors, a sequence of latent representations: $\mathbb{W} \xrightarrow{\mathbb{M}} \mathbf{z}^l = \{\mathbf{z}_1^l, \ldots, \mathbf{z}_n^l\}$ for each word $w_i$ by doing a forward pass over $s_i$. We then train a binary classifier over the representations to predict the concept $C$ minimizing the cross-entropy loss:

$$\mathcal{L}(\theta) = -\sum_i \log P_\theta(\mathbf{c}_i | \mathbf{w}_i)$$

where $P_\theta(\mathbf{c}_i | \mathbf{z}_i) = \frac{\exp(\theta_l \cdot \mathbf{z}_i)}{\sum_{c'} \exp(\theta_{l'} \cdot \mathbf{z}_i)}$ is the probability that word $\mathbf{x}_i$ is assigned concept $\mathbf{c}$. We learn the weights $\theta \in \mathbb{R}^{D \times L}$ using gradient descent. Here $D$ is the dimensionality of the latent representations $\mathbf{z}_i$ and $L$ is the size of the concept set which is 2 for a binary classifier.

## 2.4 Concept Neurons

An alternative area of research in interpreting NLP models involves conducting representation analysis at a more fine-grained level, specifically focusing on individual neurons. Our demonstration showcases how the extensive annotations of latent concepts enhance the analysis of neurons towards more intricate concepts. We show this by using a neuron ranking method called Probeless (Antverg and Belinkov, 2022) over our concept representations. The method obtains neuron rankings using an accumulative strategy, where the score of a given neuron $n$ towards a concept $C$ is defined as follows:

$$R(n, \mathcal{C}) = \mu(\mathcal{C}) - \mu(\hat{\mathcal{C}})$$

where $\mu(\mathcal{C})$ is the average of all activations $z(n, w)$, $w \in \mathcal{C}$, and $\mu(\hat{\mathcal{C}})$ is the average of activations over the random concept set. Note that the ranking for each neuron $n$ is computed independently.

# 3 Experimental Setup

**Latent Concept Data** We used a subset of the WMT News 2018 dataset, containing 250K randomly chosen sentences ($\approx$5M tokens). We set a word occurrence threshold of 10 and restricted each word type to a maximum of 10 occurrences. This selection was made to reduce computational and memory requirements when clustering high-dimensional vectors. We preserved the original embedding space to avoid information loss through dimensionality reduction techniques like PCA. Consequently, our final dataset consisted of 25,000 word types, each represented by 10 contexts.

**Concept Discovery** We apply agglomerative hierarchical clustering on contextualized feature vectors acquired through a forward pass on a pLM for the given data. The resulting representations in each layer are then clustered into 600 groups.[4]

**Concept Annotation** We used ChatGPT available through Azure OpenAI service[5] to carryout the annotations. We used a *temperature* of 0 and a *top p* value of 0.95. Setting the temperature to 0 controls the randomness in the output and produces deterministic responses.

**Pre-trained Models** Our study involved several 12-layered transformer models, including BERT-cased (Devlin et al., 2019), RoBERTa (Liu et al., 2019b), XLNet (Yang et al., 2019), and ALBERT (Lan et al., 2019) and XLM-RoBERTa (XLM-R) (Conneau et al., 2020).

**Probing and Neuron Analysis** For each annotated concept, we extract feature vectors using the relevant data. We then train linear classifiers with a categorical cross-entropy loss function, optimized using Adam (Kingma and Ba, 2014). The training process involved shuffled mini-batches of size 512 and was concluded after 10 epochs. We used a data split of 60-20-20 for train, dev, test when training classifiers. We use the same representations to obtain neuron rankings. We use NeuroX toolkit (Dalvi et al., 2023a) to train our probes and run neuron analysis.

---

[4]Dalvi et al. (2022) discovered that selecting $K$ within the range of $600 - 1000$ struck a satisfactory balance between the pitfalls of excessive clustering and insufficient clustering. Their exploration of other methods ELbow and Silhouette did not yield reliable results.

[5]https://azure.microsoft.com/en-us/products/cognitive-services/openai-service

| Q1 | Acceptable | Unacceptable |
|---|---|---|
| Majority | 244 | 25 |
| Fliess Kappa | 0.71 ("Substantial agreement") | |

| Q2 | Precise | Imprecise |
|---|---|---|
| Majority | 181 | 60 |
| Fliess Kappa | 0.34 ("Fair agreement") | |

Table 1: Inter-annotator agreement with 3 annotators. Q1: Whether the label is acceptable or unacceptable? Q2: Of the acceptable annotations how many are precise versus imprecise?

| Q3 | GPT ↑ | Equal | BCN ↑ | No Majority |
|---|---|---|---|---|
| Majority | 82 | 121 | 58 | 8 |
| Fliess Kappa | 0.56 ("Moderate agreement") | | | |

Table 2: Annotation for Q3 with 3 choices: GPT is better, labels are equivalent, human annotation is better.

## 4 Evaluation and Analysis

### 4.1 Results

To validate ChatGPT's effectiveness as an annotator, we conducted a human evaluation. Evaluators were shown a concept through a word cloud, along with sample sentences representing the concept and the corresponding GPT annotation. They were then asked the following questions:

- **Q1:** *Is the label produced by ChatGPT Acceptable or Unacceptable?* Unacceptable annotations include incorrect labels or those that ChatGPT was unable to annotate.

- **Q2:** *If a label is Acceptable, is it Precise or Imprecise?* While a label may be deemed acceptable, it may not convey the relationship between the underlying words in the concept accurately. This question aims to measure the precision of the label itself.

- **Q3:** *Is the ChatGPT label Superior or Inferior to human annotation?* BCN labels provided by Dalvi et al. (2022) are used as human annotations for this question.

In the first half of Table 1, the results indicate that 90.7% of the ChatGPT labels were considered `Acceptable`. Within the acceptable labels, 75.1% were deemed `Precise`, while 24.9% were found to be `Imprecise` (indicated by Q2 in Table 1). We also computed Fleiss' Kappa (Fleiss et al., 2013) to measure agreement among the 3 annotators. For Q1, the inter-annotator agreement was found to

| Annotation | SEM | LEX | Morph | SYN | Unint. |
|---|---|---|---|---|---|
| ChatGPT | 85.5 | 1.1 | 3.0 | X | 3.3 |
| BCN | 68.4 | 16.7 | 3.0 | 2.2 | 9.7 |

Table 3: Distribution (percentages) of concept types in ChatGPT Labels vs. Human Annotations: Semantic, Lexical, Morphological, Syntactic, Uninterpretable

be 0.71 which is considered *substantial* according to Landis and Koch (1977). However, for Q2, the agreement was 0.34 (indicating a fair level of agreement among annotators). This was expected due to the complexity and subjectivity of the task in Q2 for example annotators' knowledge and perspective on precise and imprecise labels.

**ChatGPT Labels versus Human Annotations** Next we compare the quality of ChatGPT labels to the human annotations using BERT Concept Net, a human annotated collection of latent concepts learned within the representations of BERT. BCN, however, was annotated in the form of Concept Type:Concept Sub Type (e.g., SEM:entertainment:sport:ice_hockey) unlike GPT-based annotations that are natural language descriptions (e.g. Terms related to ice hockey). Despite their lack of natural language, these reference annotations prove valuable for drawing comparative analysis between humans and ChatGPT. For Q3, we presented humans with a word cloud and three options to choose from: whether the LLM annotations are better, equailvalent, or worse than the BCN annotations. We found that ChatGPT outperformed or achieved equal performance to BCN annotations in 75.5% of cases, as shown in Table 2. The inter-annotator agreement for Q3 was found to be 0.56 which is considered *moderate*.

### 4.2 Error Analysis

The annotators identified 58 concepts where human annotated BCN labels were deemed superior. We have conducted an error analysis of these instances and will now delve into the cases where GPT did not perform well.

**Sensitive Content Models** In 10 cases, the API calls triggered one of the content policy models and failed to provide a label. The content policy models aim to prevent the dissemination of harmful, abusive, or offensive content, including hate speech, misinformation, and illegal activities. Figure 3a shows an example of a sensitive concept that

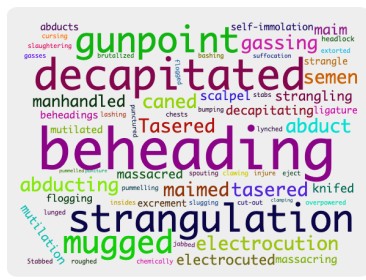

(a) Crime and Assault

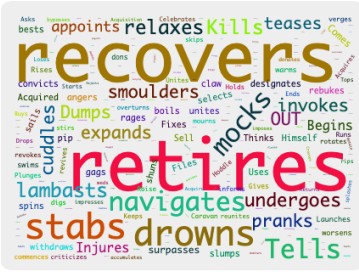

(b) 3rd Person Singular Present-tense

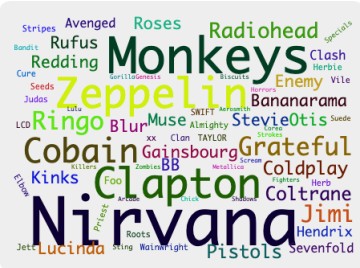

(c) Rock Bands and Artists in the US

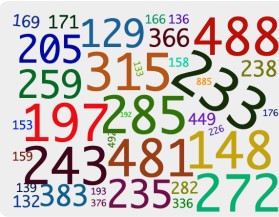

A total of `285` could prove to be rather impressive on a pitch that was turning almost from the first ball .
But there were runs at Cheltenham , where Gloucestershire finished on 315 for seven , with the only century of the entire Championship day going to Ryan Higgins , out for a 161-ball `105` .
Back in Nottingham , it was the turn of Australia 's bowlers to be eviscerated this time as Eoin Morgan 's team rewrote the record books once again by making a phenomenal `481` for six in this third one-day international .

(d) Cricket Scores

Figure 3: Failed cases for ChatGPT labeling: a) Non-labeled concepts due to LLM content policy, b) Failing to identify correct linguistic relation, c) Imprecise labeling d) Imprecise labels despite providing context

includes words related to crime and assault. This problem can be mitigated by using a version of LLM where content policy models are not enabled.

**Linguistic Ontologies**  In 8 of the concepts, human annotations (BCN) were better because the concepts were composed of words that were related through a lexical, morphological, or syntactic relationship. The default prompt we used to label the concept tends to find semantic similarity between the words, which did not exist in these concepts. For example, Figure 3b shows a concept composed of 3rd person singular present-tense verbs, but ChatGPT incorrectly labels it as Actions/Events in News Articles. However, humans are robust and can fall back to consider various linguistic ontologies.

The BCN concepts are categorized into semantic, syntactic, morphological, and lexical groups (See Table 3). As observed, both humans and ChatGPT found semantic meaning to the concept in majority of the cases. However, humans were also able to identify other linguistic relations such as lexical (e.g. grouped by a lexical property like abbreviations), morphological (e.g. grouped by the same parts-of-speech), or syntactic (e.g. grouped by position in the sentence). Note however, that prompts can be modified to capture specific linguistic property. We encourage interested readers to see our experiments on this in Appendix A.2-A.3.

**Insufficient Context**  Sometimes context contextual information is important to correctly label a concept. While human annotators (of the BCN corpus) were provided with the sentences in which the underlying words appeared, we did not provide the same to ChatGPT to keep the prompt cost-effective. However, providing context sentences in the prompt[6] along with the concept to label resulted in improved labels for 11 of the remaining 40 error cases. Figure 3d shows one such example where providing contextual information made ChatGPT to correctly label the concept as Cricket Scores as opposed to Numerical Data the label that it gives without seeing contextual information. However, providing context information didn't consistently prove helpful. Figure 3c shows a concept, where providing contextual information did not result in the accurate label: Rock Bands and Artists in the US, as identified by the humans.

**Uninterpretable Concepts**  Conversely, we also annotated concepts that were considered uninterpretable or non-meaningful by the human annotators in the BCN corpus and in 21 out 26 cases, ChatGPT accurately assigned labels to these concepts. The proficiency of ChatGPT in processing extensive textual data enables it to provide accurate labels for these concepts.

---

[6]We gave 10 context sentences to ChatGPT.

| tag | Label | ALBERT | XLNet |
|-----|-------|--------|-------|
| c301 | Gender-related Nouns and pronouns | 0.95 | 0.86 |
| c533 | LGBTQ+ | 0.97 | 0.97 |
| c439 | Sports commentary terms | 0.91 | 0.81 |
| c173 | Football team names and stadiums | 0.96 | 0.94 |
| c348 | Female names and titles | 0.98 | 0.94 |
| c149 | Tennis players' names | 0.95 | 0.92 |
| c487 | Spanish Male Names | 0.96 | 0.94 |
| c564 | Cities and Universities in southeastern US | 0.97 | 0.90 |
| c263 | Locations in New York City | 0.95 | 0.92 |
| c247 | Scandinavian/Nordic names and places | 0.98 | 0.95 |
| c438 | Verbs for various actions and outcomes | 0.94 | 0.87 |
| c44 | Southeast Asian Politics and Ethnic Conflict | 0.97 | 0.94 |
| c421 | Names of people and places in the middle east | 0.94 | 0.95 |
| c245 | Middle East conflict | 1.00 | 0.93 |
| c553 | Islamic terminology | 0.96 | 0.89 |
| c365 | Criminal activities | 0.93 | 0.89 |
| c128 | Medical and Healthcare terminology | 0.98 | 0.95 |

Table 4: Using latent concepts to make cross-model comparison using Probing Classifiers

# 5 Concept-based Interpretation Analysis

Now that we have established the capability of large language models like ChatGPT in providing rich semantic annotations, we will showcase how these annotations can facilitate extensive fine-grained analysis on a large scale.

## 5.1 Probing Classifiers

Probing classifiers is among the earlier techniques used for interpretability, aimed at examining the knowledge encapsulated in learned representations. However, their application is constrained by the availability of supervised annotations, which often focus on conventional linguistic knowledge and are subject to inherent limitations (Hewitt and Liang, 2019). We demonstrate that using GPT-based annotation of latent concepts learned within these models enables a direct application towards fine-grained probing analysis. By annotating the latent space of five renowned pre-trained language models (pLMs): BERT, ALBERT, XLM-R, XLNet, and RoBERTa – we developed a comprehensive Transformers Concept Net. This net encompasses 39,000 labeled concepts, facilitating cross-architectural comparisons among the models. Table 4 showcases a subset[7] of results comparing ALBERT and XLNet through probing classifiers.

We can see that the model learns concepts that may not directly align with the pre-defined human onotology. For example, it learns a concept based on Spanish Male Names or Football team names and stadiums. Identifying how

[7] For a larger sample of concepts and additional models, please refer to Appendix B.

fine-grained concepts are encoded within the latent space of a model enable applications beyond interpretation analysis. For example it has direct application in model editing (Meng et al., 2023) which first trace where the model store any concept and then change the relevant parameters to modify its behavior. Moreover, identifying concepts that are associated with gender (e.g., Female names and titles), religion (e.g. Islamic Terminology), and ethnicity (e.g., Nordic names) can aid in elucidating the biases present in these models.

## 5.2 Neuron Analysis

Neuron analysis examines the individual neurons or groups of neurons within neural NLP models to gain insights into how the model represents linguistic knowledge. However, similar to general interpretability, previous studies in neuron analysis are also constrained by human-in-the-loop (Karpathy et al., 2015; Kádár et al., 2017) or pre-defined linguistic knowledge (Lakretz et al., 2019; Dalvi et al., 2019; Hennigen et al., 2020). Consequently, the resulting neuron explanations are subject to the same limitations we address in this study.

Our work demonstrates that annotating the latent space enables neuron analysis of intricate linguistic hierarchies learned within these models. For example, Dalvi et al. (2019) and Hennigen et al. (2020) only carried out analysis using very coarse morphological categories (e.g. adverbs, nouns etc.) in parts-of-speech tags. We now showcase how our discovery and annotations of fine-grained latent concepts leads to a deeper neuron analysis of these models. In our analysis of BERT-based part-of-speech tagging model, we discovered 17 fine-grained concepts of adverb (in the final layer). It is evident that BERT learns a highly detailed semantic hierarchy, as maintains separate concepts for the adverbs of frequency (e.g., "rarely, sometimes") versus adverbs of manner (e.g., "quickly, softly"). We employed the *Probeless* method (Antverg and Belinkov, 2022) to search for neurons associated with specific kinds of adverbs. We also create a super adverb concept encompassing all types of adverbs, serving as the overarching and generic representation for this linguistic category and obtain neurons associated with the concept. We then compare the neuron ranking obtained from the super concept to the individual rankings from sub concepts. Interestingly, our findings revealed that the top-ranking neurons responsible for learning

| Super Concept | # Sub Concepts | Alignment |
|---|---|---|
| Adverbs | 17 | 0.36 |
| ↪ c155: Frequency and manner | | 0.30 |
| ↪ c136: Degree/Intensity | | 0.30 |
| ↪ c057: Frequency | | 0.40 |
| Nouns | 13 | 0.28 |
| ↪ c231: Activities and Objects | | 0.60 |
| ↪ c279: Industries/Sectors | | 0.60 |
| ↪ c440: Professions | | 0.10 |
| Adjectives | 17 | 0.21 |
| ↪ c299: Product Attributes | | 0.30 |
| ↪ c053: Comparative Adjectives | | 0.30 |
| ↪ c128: Quality/Appropriateness | | 0.40 |
| Numbers | 17 | 0.23 |
| ↪ c549: Prices | | 0.50 |
| ↪ c080: Quantities | | 0.10 |
| ↪ c593: Monetary Values | | 0.10 |

Table 5: Neuron Analysis on *Super Concepts* extracted from BERT-base-cased-POS model. The alignment column shows the intersection between the top 10 neurons in the Super concept and the Sub concepts. For detailed results please check Appendix C (See Table 11)

Figure 4: Neuron overlap between an Adverb Super Concept and sub concepts. Sub concepts shown are Adverbs of frequency and manner (c155), Adverbs of degree/intensity (c136), Adverbs of Probability and Certainty (c265), Adverbs of Frequency (c57), Adverbs of manner and opinion (c332), Adverbs of preference/choice (c570), Adverbs indicating degree or extent (c244), Adverbs of Time (c222).

the super concept are often distributed among the top neurons associated with specialized concepts, as shown in Figure 4 for adverbial concepts. The results, presented in Table 5, include the number of discovered sub concepts in the column labeled # Sub Concepts and the Alignment column indicates the percentage of overlap in the top 10 neurons between the super and sub concepts for each specific adverb concept. The average alignment across all sub concepts is indicated next to the super concept. This observation held consistently across various properties (e.g. Nouns, Adjectives and Numbers) as shown in Table 5. For further details please refer to Appendix C).

Note that previously, we couldn't identify neurons with such specific explanations, like distinguishing neurons for numbers related to currency values from those for years of birth or neurons differentiating between cricket and hockey-related terms. Our large scale concept annotation enables locating neurons that capture the fine-grained aspects of a concept. This enables applications such as manipulating network's behavior in relation to that concept. For instance, Bau et al. (2019) identified "tense" neurons within Neural Machine Translation (NMT) models and successfully changed the output from past to present tense by modifying the activation of these specific neurons. However, their study was restricted to very few coarse concepts for which annotations were available.

## 6 Related Work

With the ever-evolving capabilities of the LLMs, researchers are actively exploring innovative ways to harness their assistance. Prompt engineering, the process of crafting instructions to guide the behavior and extract relevant knowledge from these oracles, has emerged as a new area of research (Lester et al., 2021; Liu et al., 2021; Kojima et al., 2023; Abdelali et al., 2023; Dalvi et al., 2023b). Recent work has established LLMs as highly proficient annotators. Ding et al. (2022) carried out evaluation of GPT-3's performance as a data annotator for text classification and named entity recognition tasks, employing three primary methodologies to assess its effectiveness. Wang et al. (2021) showed that GPT-3 as an annotator can reduce cost from 50-96% compared to human annotations on 9 NLP tasks. They also showed that models trained using GPT-3 labeled data outperformed the GPT-3 few-shot learner. Similarly, Gilardi et al. (2023) showed that ChatGPT achieves higher zero-shot accuracy compared to crowd-source workers in various annotation tasks, encompassing relevance, stance, topics, and frames detection. Our work is different from previous work done using GPT as annotator. We annotate the latent concepts encoded within the embedding space of pre-trained language models. We demonstrate how such a large scale annotation enriches representation analysis via application in probing classifiers and neuron analysis.

# 7 Conclusion

The scope of previous studies in interpreting neural language models is limited to general ontologies or small-scale manually labeled concepts. In our research, we showcase the effectiveness of Large Language Models, specifically ChatGPT, as a valuable tool for annotating latent spaces in pre-trained language models. This large-scale annotation of latent concepts broadens the scope of interpretation from human-defined ontologies to encompass all concepts learned within the model, and eliminates the human-in-the-loop effort for annotating these concepts. We release a comprehensive GPT-annotated Transformers Concept Net (TCN) consisting of 39,000 concepts, extracted from a wide range of transformer language models. TCN empowers the researchers to carry out large-scale interpretation studies of these models. To demonstrate this, we employ two widely used techniques in the field of interpretability: probing classifiers and neuron analysis. This novel dimension of analysis, previously absent in earlier studies, sheds light on intricate aspects of these models. By showcasing the superiority, adaptability, and diverse applications of ChatGPT annotations, we lay the groundwork for a more comprehensive understanding of NLP models.

## Limitations

We list below limitations of our work:

- While it has been demonstrated that LLMs significantly reduce the cost of annotations, the computational requirements and response latency can still become a significant challenge when dealing with extensive or high-throughput annotation pipeline like ours. In some cases it is important to provide contextual information along with the concept to obtain an accurate annotation, causing the cost go up. Nevertheless, this is a one time cost for any specific model, and there is optimism that future LLMs will become more cost-effective to run.

- Existing LLMs are deployed with content policy filters aimed at preventing the dissemination of harmful, abusive, or offensive content. However, this limitation prevents the models from effectively labeling concepts that reveal sensitive information, such as cultural and racial biases learned within the model to be interpreted. For example, we were unable to extract a label for

racial slurs in the hate speech detection task. This restricts our concept annotation approach to only tasks that are not sensitive to the content policy.

- The information in the world is evolving, and LLMs will require continuous updates to reflect the accurate state of the world. This may pose a challenge for some problems (e.g. news summarization task) where the model needs to reflect an updated state of the world.

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

## Appendix

## A Prompts

### A.1 Optimal Prompt

Initially, we used a simple prompt to ask the model to provide labels for a list of words keeping the system description unchanged:

```
Assistant is a large language model
trained by OpenAI
```
Prompt Body: `Give the following list of words a short label: ["word 1", "word 2", ..., "word N"]`

The output format from the first prompt was unclear as it included illustrations, which was not our intention. After multiple design iterations, we developed a prompt that returned the labels in the desired format. In this revised prompt, we modified the system description as follows:

```
Assistant is a large language model
trained by OpenAI.
```
Instructions: `When asked for labels, only the labels and nothing else should be returned.`

We also modified the prompt body to:

```
Give a short and concise label that
best describes the following list of
words: ["word 1", "word 2", ..., "word
N"]
```

Figure 5 shows some sample concepts learned in the last layer of BERT-base-cased along with their labels.

### A.2 Prompts For Lexical Concepts

During the error analysis (Section 4.2), we discovered that GPT struggled to accurately label concepts composed of words sharing a lexical property, such as a common suffix. However, we were able to devise a solution to address this issue by curating the prompt to effectively label such concepts. We modified the prompt to identify concepts that contain common n-grams.

```
Give a short and concise label
describing the common ngrams between the
words of the given list
```
Note: `Only one common ngram should be returned. If there is no common ngram reply with 'NA'`

Using this improved we were able to correct 100% of the labeling errors in the concepts having lexical coherence. See Figure 7a for example. With the default prompt it was labelled as `Superlative and ordinal adjectives` and with the modified prompt, it was labeled as `Hyphenated, cased & -based suffix`.

### A.3 Prompts for POS Concepts

Similarly we were able to modify the prompt to correctly label concepts that were made from words having common parts-of-speech. From the prompts we tested, the best performing one is below:

```
Give a short and concise label
describing the common part of speech tag
between the words of the given list
```
Note: `The part of speech tag should be chosen from the Penn Treebank. If there's no common part of speech tag reply with 'NA'`

In Figure 7b, we present an example of a concept labeled as `Surnames with 'Mc' prefix`. However, it is important to note that not all the names in this concept actually begin with the "Mc" prefix. The appropriate label for this concept would be NNP: Proper Nouns or SEM: Irish Names. With the POS-based prompt, we are able to achieve the former.

### A.4 Providing Context

Our analysis revealed that including contextual information is crucial for accurately labeling concepts in certain cases. As shown in Figure 8, concepts were incorrectly labeled as `Numerical Data` despite representing different entities. Incorporating context enables us to obtain more specific labels. However, we face limitations in the number of input tokens we can provide to the model, which impacts the quality of the labels. Using context of 10 sentences we were able to correct 9 of the 38 erroneous labels.

### A.5 Other Details

**Tokens Versus Types** We observed that the quality of labels is influenced by the word frequency in the given list. Using tokens instead of types leads to more meaningful labels. However, when the latent concept includes hate speech words, passing a token list results in failed requests due to content policy violations. In such cases, we opted to pass the list of types instead. Although this mitigates the issue to a certain extent, it does not completely

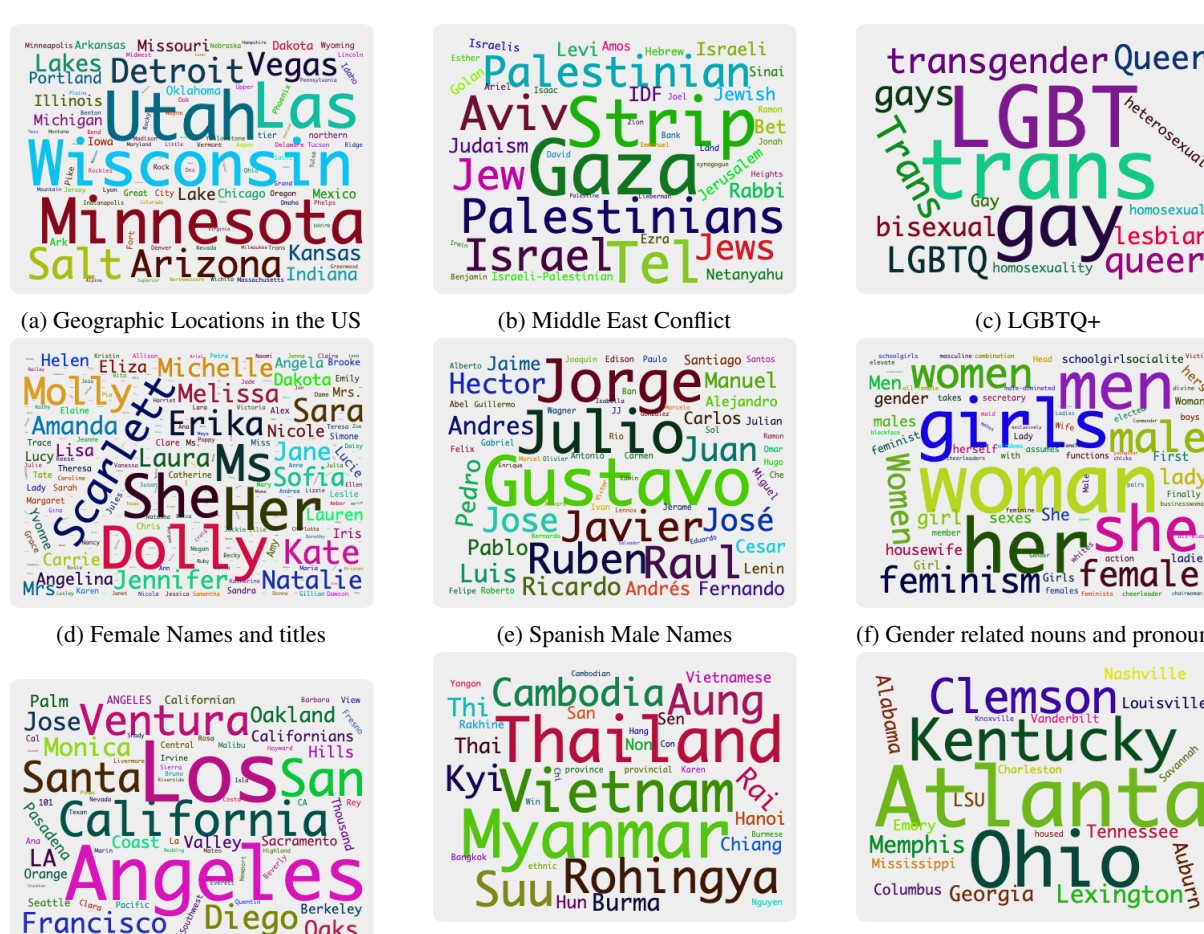

(a) Geographic Locations in the US

(b) Middle East Conflict

(c) LGBTQ+

(d) Female Names and titles

(e) Spanish Male Names

(f) Gender related nouns and pronouns

(g) Geographic Locations in California

(h) SE Asian Politics and Ethnic Conflict

(i) List of cities and universities in southeastern US

Figure 5: Sample Concepts Learned in the last layer of BERT

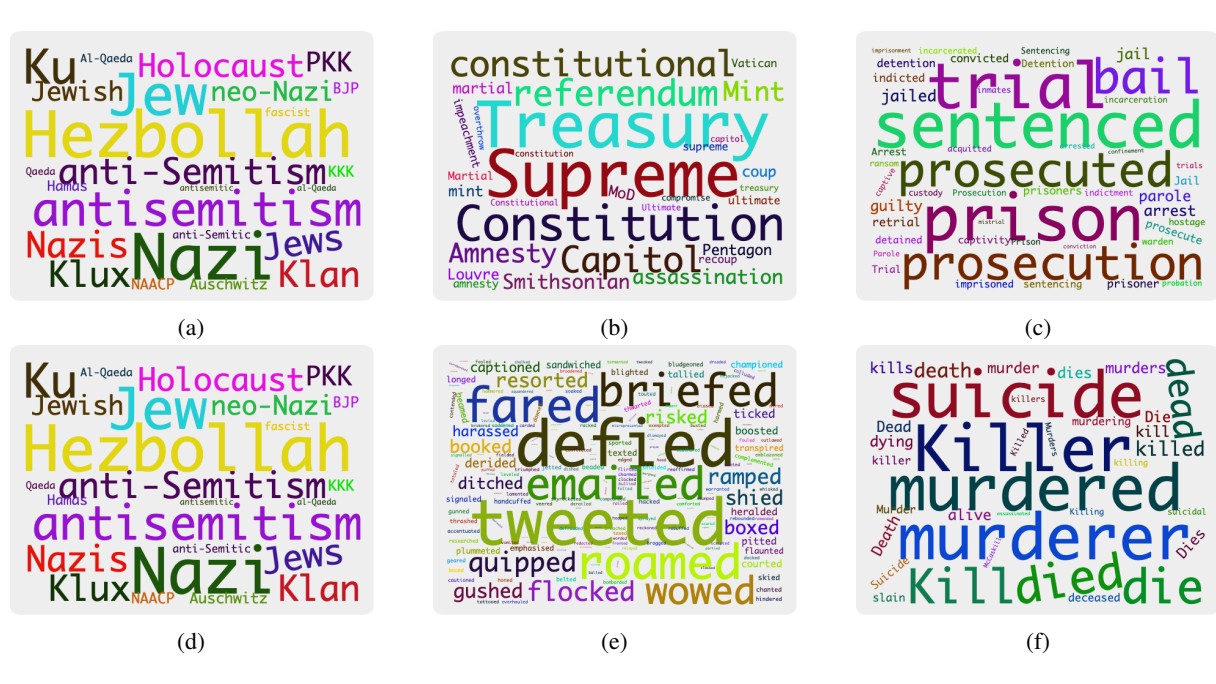

(a)

(b)

(c)

(d)

(e)

(f)

Figure 6: Failed Requests in Albert

| Tag | Human Label | 1 Label Response | 3-Keyword Response |
|---|---|---|---|
| c533 | LEX:hyphenated | Superlative and ordinal adjectives. | ['second', 'highest', 'biggest'] |
| c84 | LEX:hyphenated | Accomplishments and Awards | ['Award-winning', 'Nominated', 'Multi-time'] |
| c783 | LEX:hyphenated | Sports scores and point differentials. | ['points', 'wins', 'scores'] |
| c621 | LEX:hyphenated | Describing people's relationships and family status. | [family, relationships, parenthood] |
| c869 | LEX:hyphenated | Tennis Scores. | ['Tennis', 'Scores', 'Games'] |
| c833 | LEX:hyphenated | Location-based adjectives | [based, area, listed] |
| c588 | LEX:hyphenated | US political party affiliations by state and district. | [Republican, Democrat, State Abbreviations] |
| c639 | LEX:hyphenated | Football scores. | ['Scores', 'Football', 'Winning'] |
| c934 | LEX:hyphenated | Sports scores. | ['scores', 'victories', 'defeats'] |
| c850 | LEX:case:title_case | Philippine Places and Names | ['Philippines', 'Tourism', 'Volcano'] |
| c286 | LEX:case:title_case | List of surnames. | [Last names, English, List] |
| c982 | LEX:case:title_case | Sports-related terms. | ['Football', 'Sports', 'Legends'] |
| c231 | SYN:position:first_word | Sports Terminology | ['Footballers', 'Tries', 'Substitutes'] |
| c784 | SYN:position:first_word | Numerical data. | [Numbers, Decimals, List] |
| c728 | SYN:position:first_word | Action-oriented verbs and adjectives. | [Improving, Ensuring, Capturing] |
| c672 | SYN:position:first_word | Verbs describing actions and states. | [Fluent, Struggling, Showcasing] |
| c886 | LEX:case:title_case | Describing communication actions. | [Referring, Recalling, Revealing] |
| c865 | LEX:case:title_case | Baseball player names. | [Bregman, Scherzer, Puig] |
| c734 | LEX:case:title_case | Island names. | ['Islands', 'Caribbean', 'Indian Ocean'] |
| c818 | LEX:case:title_case | Ethnicities and Cities in the Balkans | ['Bosnian', 'Albanian', 'Yugoslavia'] |

Table 6: Prompting ChatGPT to label a concept with keywords instead of one label

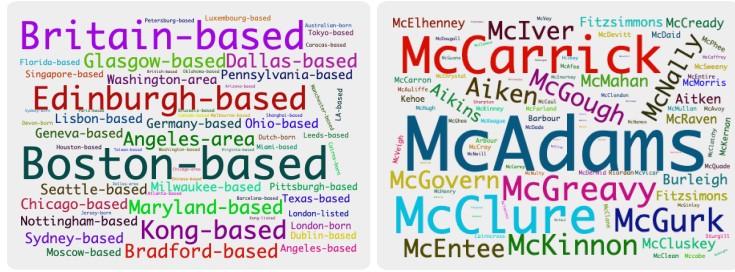

(a) Hyphenated,cased & -based suffix   (b) NNP: Proper Nouns

Figure 7: Illustrating lexical and POS concepts: (a) A concept that exhibits multiple lexical properties, such as being hyphenated and cased. ChatGPT assigns a label based on the shared "-based" ngram found among most words in the cluster. (b) ChatGPT labeled this concept as NNP (proper noun)

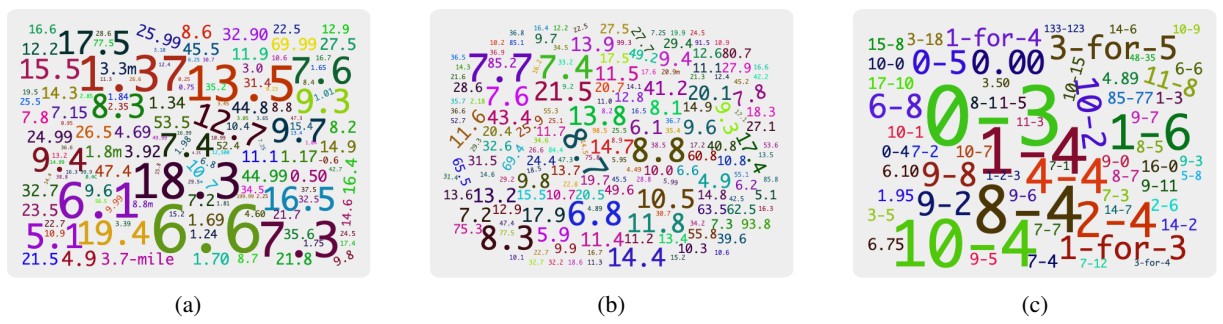

(a)          (b)          (c)

Figure 8: Highlighting the Significance of Context: (a) Money Figures (b) Percentages (c) Baseball Scores. All of these concepts were mislabeled as Numerical values by ChatGPT. Providing the context sentences we are able to obtain the correct label

resolve it. Refer to Figure 6 for examples of failed requests with Albert.

**Keyword prompts**   We also explored prompts to return 3 keywords that describe the concept instead of returning a concise label in an effort to produce multiple labels like BCN.
```
Instructions:
When asked for keywords, only the
keywords and nothing else should be
returned.
If asked for 3 keywords, the keywords
should be returned in the form of
[keyword_1, keyword_2, keyword_3]
```

To ensure compliance with our desired output format, we introduced a second instruction since the model was not following the first instruction as intended. We also modified the prompt body to:

```
Give 3 keywords that best describe the
following list of words
```

Unfortunately, this prompt did not provide accurate labels, as illustrated in Table 6.

## B   Probing Classifiers

### B.1   Running Probes At Scale

**Probing For Fine-grained Semantic Concepts** We used the NeuroX toolkit to train a linear probe for several concepts chosen from layers 3, 9 and 12 of BERT-base-cased. We used a train/val/test splits of 0.6, 0.2, 0.2 respectively. Tables 8 and 9 show the data statistics and the probe results respectively. Table 10 shows results of probes trained on concepts chosen from multiple layers of ALBERT. In Table 7 we carried out a cross architectural comparison across the models by training probes towards the same set of concepts.

## C   Neuron Analysis Results

**Neurons Associated with POS concepts**   We performed an annotation process on the final layer of a fine-tuned version of BERT-base-cased, specifically focusing on the task of parts-of-speech tagging. Once we obtained the labels, we organized them into super concepts based on a shared characteristic among smaller concepts. For instance, we grouped together various concepts labeled as nouns, as well as concepts representing adjectives, adverbs, and numerical data. To assess the alignment between the sub concepts and the super concept, we calculated the occurrence percentage of the top 10 neurons from the sub concept within the top 10 neurons of the super concept. The outcomes of this analysis can be found in table 11, illustrating the average alignment between the sub concepts and the super concepts.

**Neurons Associated with the Names concepts** We replicated the experiment using named entity concepts derived from the final layer of bert-base-cased. The findings are presented in table 12.

| tag | Label | BERT | Sel | ALBERT | Sel | XLNet | Sel | XLM-R | Sel | RoBERTa | Sel |
|---|---|---|---|---|---|---|---|---|---|---|---|
| c301 | Gender-related Nouns and pronouns | 0.98 | 0.16 | 0.95 | 0.14 | 0.86 | 0.24 | 0.94 | 0.23 | 0.95 | 0.26 |
| c533 | LGBTQ+ | 1 | 0.18 | 0.97 | 0.33 | 0.97 | 0.43 | 1 | 0.25 | 1 | 0.14 |
| c439 | Sports commentary terms | 0.94 | 0.2 | 0.91 | 0.18 | 0.81 | 0.05 | 0.87 | 0.11 | 0.86 | 0.09 |
| c173 | Football team names and stadiums | 0.94 | 0.2 | 0.96 | 0.27 | 0.94 | 0.24 | 0.95 | 0.2 | 0.97 | 0.34 |
| c348 | Female names and titles | 0.98 | 0.29 | 0.98 | 0.29 | 0.94 | 0.21 | 0.96 | 0.16 | 0.97 | 0.24 |
| c149 | Tennis players' names | 0.98 | 0.27 | 0.95 | 0.25 | 0.92 | 0.19 | 0.92 | 0.17 | 0.92 | 0.19 |
| c487 | Spanish Male Names | 0.95 | 0.26 | 0.96 | 0.07 | 0.94 | 0.37 | 0.91 | 0.25 | 0.98 | 0.28 |
| c564 | Cities and Universities in southeastern US | 0.97 | 0.12 | 0.97 | 0.11 | 0.9 | 0.18 | 0.97 | 0.29 | 0.96 | 0.22 |
| c263 | Locations in New York City | 0.95 | 0.25 | 0.95 | 0.22 | 0.92 | 0.26 | 0.95 | 0.26 | 0.95 | 0.17 |
| c247 | Scandinavian/Nordic names and places | 0.97 | 0.22 | 0.98 | 0.27 | 0.95 | 0.29 | 0.96 | 0.21 | 0.98 | 0.29 |
| c438 | Verbs for various actions and outcomes | 0.97 | 0.12 | 0.94 | 0.09 | 0.87 | 0.23 | 0.92 | 0.11 | 0.92 | 0.14 |
| c44 | Southeast Asian Politics and Ethnic Conflict | 0.97 | 0.17 | 0.97 | 0.19 | 0.94 | 0.25 | 0.93 | 0.09 | 0.95 | 0.16 |
| c421 | Names of people and places in the middle east | 0.97 | 0.06 | 0.94 | 0.28 | 0.95 | 0.22 | 0.93 | 0.31 | 0.92 | 0.12 |
| c245 | Middle East conflict | 0.98 | 0.26 | 1 | 0.25 | 0.93 | 0.29 | 0.93 | 0.25 | 0.95 | 0.22 |
| c553 | Islamic terminology | 1 | 0.15 | 0.96 | 0.4 | 0.89 | 0.29 | 0.89 | 0.16 | 0.95 | 0.26 |
| c365 | Criminal activities | 0.97 | 0.15 | 0.93 | 0.17 | 0.89 | 0.35 | 0.9 | 0.15 | 0.93 | 0.21 |
| c128 | Medical and Healthcare terminology | 0.98 | 0.17 | 0.98 | 0.21 | 0.95 | 0.15 | 0.94 | 0.24 | 0.95 | 0.27 |

Table 7: Training Probes towards latent concepts discovered in various Models. Reporting classifier accuracy on test-set along with respective selectivity numbers

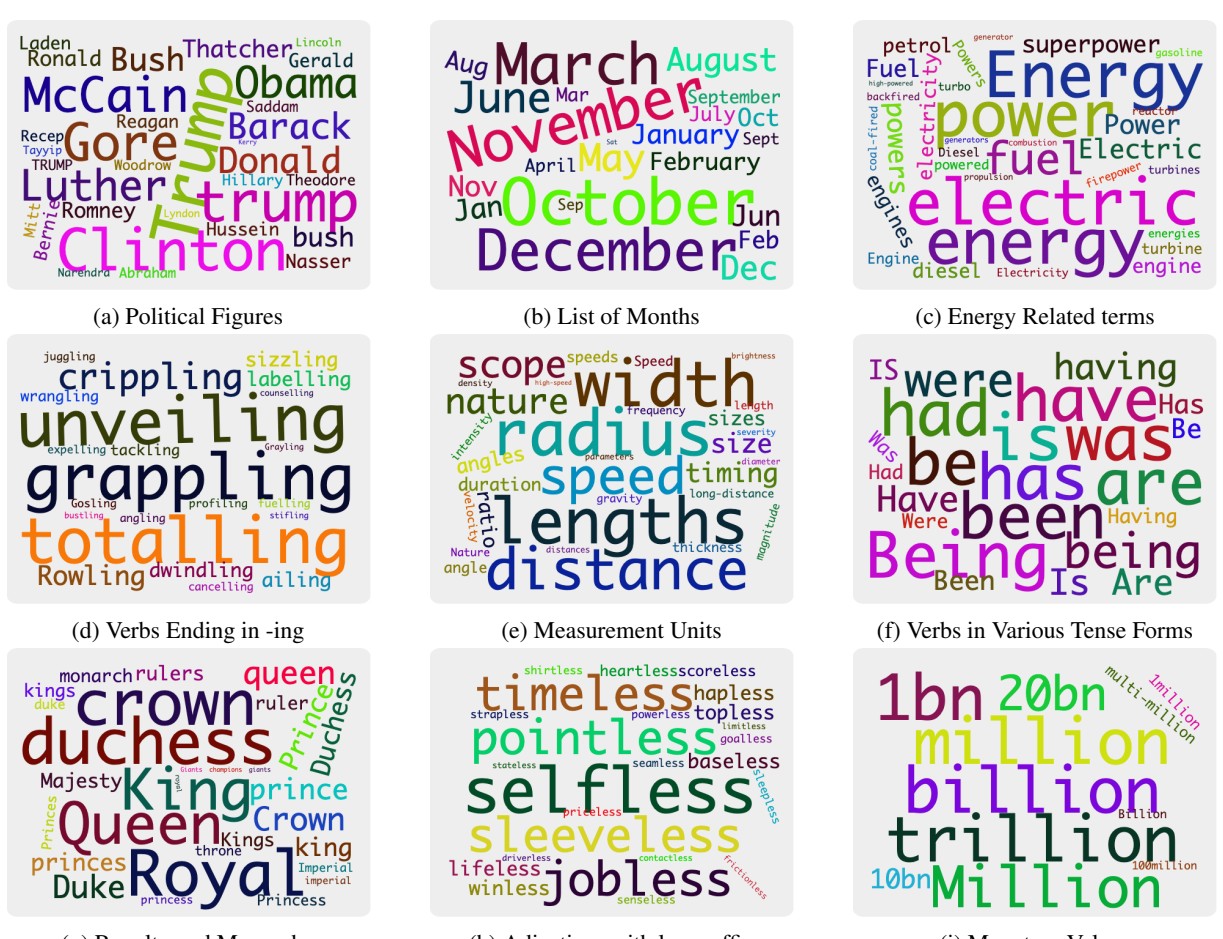

(a) Political Figures    (b) List of Months    (c) Energy Related terms

(d) Verbs Ending in -ing    (e) Measurement Units    (f) Verbs in Various Tense Forms

(g) Royalty and Monarchy    (h) Adjectives with less suffix    (i) Monetary Values

Figure 9: Sample Concepts learned in the ALBERT Model

| Layer | Tag | Label | Tokens | Types | Sents | Train | Val | Test |
|---|---|---|---|---|---|---|---|---|
| 3 | c90 | Financial terms. | 220 | 22 | 214 | 285 | 95 | 96 |
| 3 | c336 | Photography-related terms. | 290 | 29 | 273 | 388 | 130 | 130 |
| 3 | c112 | Middle Eastern Conflict | 620 | 62 | 523 | 992 | 331 | 331 |
| 3 | c506 | Diversity and Ethnicity. | 240 | 24 | 225 | 331 | 111 | 112 |
| 3 | c390 | List of surnames. | 4298 | 430 | 4049 | 5530 | 1844 | 1844 |
| 3 | c331 | Emotions/Feelings. | 400 | 40 | 396 | 484 | 162 | 162 |
| 3 | c592 | Animal names. | 220 | 22 | 208 | 268 | 90 | 90 |
| 3 | c25 | Keywords related to discrimination and inequality. | 340 | 34 | 325 | 440 | 147 | 147 |
| 3 | c500 | List of female names. | 2913 | 292 | 2735 | 3867 | 1289 | 1290 |
| 3 | c414 | Healthcare | 510 | 51 | 475 | 752 | 251 | 251 |
| 3 | c31 | List of male first names. | 1130 | 113 | 1078 | 1422 | 474 | 474 |
| 3 | c173 | Animals | 760 | 76 | 704 | 994 | 332 | 332 |
| 3 | c72 | Natural Disasters and Weather Events | 701 | 71 | 635 | 1022 | 341 | 341 |
| 3 | c514 | English counties | 297 | 30 | 286 | 373 | 124 | 125 |
| 3 | c178 | Body Parts | 430 | 43 | 405 | 588 | 196 | 196 |
| 3 | c340 | Media and Journalism. | 379 | 38 | 365 | 518 | 173 | 173 |
| 3 | c432 | Power and Status. | 310 | 31 | 306 | 385 | 128 | 129 |
| 3 | c8 | Verbs | 1028 | 103 | 1018 | 1243 | 414 | 415 |
| 3 | c408 | -Verbs ending in "-ing" | 510 | 51 | 504 | 615 | 205 | 206 |
| 3 | c479 | City names | 130 | 13 | 127 | 159 | 53 | 54 |
| 3 | c343 | Surnames | 490 | 49 | 464 | 613 | 204 | 205 |
| 3 | c577 | Disability-related terms. | 140 | 14 | 133 | 172 | 58 | 58 |
| 9 | c26 | Negative sentiment. | 798 | 118 | 782 | 1036 | 346 | 346 |
| 9 | c122 | Security Measures | 457 | 70 | 446 | 584 | 195 | 195 |
| 9 | c423 | Label: Islamic Extremism/Terrorism | 248 | 30 | 222 | 357 | 119 | 120 |
| 9 | c299 | Middle Eastern and North African countries and cities | 531 | 57 | 460 | 844 | 282 | 282 |
| 9 | c192 | Diversity and Identity | 314 | 50 | 279 | 506 | 169 | 169 |
| 9 | c468 | Russian male names. | 125 | 18 | 123 | 153 | 51 | 52 |
| 9 | c588 | Gender-related terms. | 161 | 19 | 146 | 236 | 79 | 79 |
| 9 | c74 | Financial terms | 672 | 96 | 607 | 1118 | 373 | 373 |
| 9 | c503 | Middle East Conflict. | 230 | 27 | 185 | 404 | 135 | 135 |
| 9 | c325 | Violent Crimes | 292 | 60 | 287 | 386 | 129 | 129 |
| 9 | c535 | Academic Research. | 233 | 26 | 227 | 332 | 111 | 111 |
| 9 | c256 | List of names | 1069 | 149 | 1026 | 1375 | 458 | 459 |
| 9 | c507 | Positive Adjectives | 389 | 69 | 380 | 505 | 168 | 169 |
| 9 | c345 | List of Chinese surnames. | 407 | 65 | 378 | 567 | 189 | 190 |
| 12 | c259 | List of names | 223 | 174 | 221 | 273 | 91 | 92 |
| 12 | c62 | Adverbs | 1221 | 351 | 1133 | 3769 | 1256 | 1257 |
| 12 | c128 | Medical and Healthcare Terminology. | 395 | 70 | 369 | 662 | 221 | 221 |
| 12 | c301 | Gender-related nouns and pronouns. | 418 | 74 | 377 | 883 | 294 | 295 |
| 12 | c37 | List of male names. | 872 | 372 | 807 | 1460 | 487 | 487 |
| 12 | c281 | Adverbs | 928 | 264 | 927 | 1178 | 393 | 393 |
| 12 | c220 | List of surnames. | 3886 | 832 | 3652 | 6378 | 2126 | 2126 |
| 12 | c432 | List of Male Names | 279 | 159 | 227 | 474 | 158 | 158 |
| 12 | c439 | Sports commentary terms. | 250 | 181 | 189 | 687 | 229 | 230 |
| 12 | c173 | List of football team names and stadiums. | 373 | 81 | 287 | 849 | 283 | 284 |
| 12 | c348 | List of female names and titles. | 575 | 301 | 571 | 774 | 258 | 258 |
| 12 | c142 | Conflict and War | 407 | 106 | 385 | 582 | 194 | 194 |
| 12 | c245 | Middle East Conflict | 249 | 42 | 196 | 453 | 151 | 152 |
| 12 | c210 | List of male first names. | 317 | 205 | 268 | 470 | 157 | 157 |
| 12 | c564 | List of cities and universities in the southeastern United States. | 175 | 21 | 162 | 229 | 76 | 77 |
| 12 | c533 | LGBTQ+ | 131 | 15 | 118 | 188 | 63 | 63 |
| 12 | c19 | Complex relationships and interactions between family members and partners. | 346 | 56 | 333 | 546 | 182 | 182 |
| 12 | c263 | Locations in New York City | 205 | 48 | 186 | 386 | 129 | 129 |
| 12 | c487 | List of Spanish male names. | 184 | 63 | 174 | 242 | 81 | 81 |
| 12 | c247 | Scandinavian/Nordic names and places. | 334 | 64 | 305 | 502 | 168 | 168 |
| 12 | c44 | Southeast Asian Politics and Ethnic Conflict | 210 | 33 | 149 | 332 | 111 | 111 |
| 12 | c438 | Verbs for various actions and outcomes. | 896 | 377 | 847 | 1600 | 534 | 534 |
| 12 | c421 | Names of people and places in the Middle East | 270 | 48 | 230 | 361 | 120 | 121 |
| 12 | c553 | Islamic Terminology. | 164 | 26 | 146 | 253 | 84 | 85 |
| 12 | c149 | List of tennis players' names. | 238 | 82 | 183 | 394 | 132 | 132 |
| 12 | c365 | Criminal activities | 365 | 88 | 337 | 496 | 166 | 166 |

Table 8: Statistics for concepts extracted from Bert-base-cased and the training, dev, test splits used to train the classifier

| Layer | Tag | Label | val acc | val C acc | sel val | test acc | test c acc | sel test |
|---|---|---|---|---|---|---|---|---|
| 3 | c90 | Financial terms. | 0.98 | 0.65 | 0.33 | 0.98 | 0.78 | 0.20 |
| 3 | c336 | Photography-related terms. | 0.99 | 0.74 | 0.25 | 1 | 0.76 | 0.24 |
| 3 | c112 | Middle Eastern Conflict | 0.99 | 0.89 | 0.10 | 1 | 0.86 | 0.14 |
| 3 | c506 | Diversity and Ethnicity. | 1 | 0.78 | 0.22 | 0.98 | 0.75 | 0.23 |
| 3 | c390 | List of surnames. | 0.97 | 0.82 | 0.15 | 0.97 | 0.82 | 0.15 |
| 3 | c331 | Emotions/Feelings. | 0.98 | 0.82 | 0.16 | 0.99 | 0.78 | 0.21 |
| 3 | c592 | Animal names. | 1 | 0.68 | 0.32 | 1 | 0.73 | 0.27 |
| 3 | c25 | Keywords related to discrimination and inequality. | 0.99 | 0.81 | 0.18 | 0.98 | 0.77 | 0.21 |
| 3 | c500 | List of female names. | 0.98 | 0.82 | 0.16 | 0.99 | 0.83 | 0.16 |
| 3 | c414 | Healthcare | 1 | 0.77 | 0.23 | 1 | 0.79 | 0.21 |
| 3 | c31 | List of male first names. | 0.99 | 0.85 | 0.14 | 1 | 0.83 | 0.17 |
| 3 | c173 | Animals | 0.99 | 0.78 | 0.21 | 0.99 | 0.75 | 0.24 |
| 3 | c72 | Natural Disasters and Weather Events | 0.99 | 0.80 | 0.19 | 0.99 | 0.78 | 0.21 |
| 3 | c514 | English counties | 1 | 0.74 | 0.26 | 1 | 0.76 | 0.24 |
| 3 | c178 | Body Parts | 0.99 | 0.84 | 0.15 | 0.98 | 0.89 | 0.9 |
| 3 | c340 | Media and Journalism. | 0.98 | 0.76 | 0.22 | 1 | 0.78 | 0.22 |
| 3 | c432 | Power and Status. | 0.99 | 0.79 | 0.20 | 1 | 0.78 | 0.22 |
| 3 | c8 | Verbs | 0.99 | 0.88 | 0.11 | 0.99 | 0.89 | 0.10 |
| 3 | c408 | -Verbs ending in "-ing" | 1 | 0.68 | 0.32 | 1 | 0.73 | 0.27 |
| 3 | c479 | City names | 0.98 | 0.68 | 0.30 | 1 | 0.83 | 0.17 |
| 3 | c343 | Surnames | 1 | 0.74 | 0.26 | 0.98 | 0.74 | 0.24 |
| 3 | c577 | Disability-related terms. | 1 | 0.82 | 0.18 | 1 | 0.78 | 0.22 |
| 9 | c26 | Negative sentiment. | 0.98 | 0.79 | 0.19 | 0.99 | 0.8 | 0.19 |
| 9 | c122 | Security Measures | 0.98 | 0.81 | 0.17 | 0.99 | 0.82 | 0.17 |
| 9 | c423 | Label: Islamic Extremism/Terrorism | 1 | 0.77 | 0.23 | 1 | 0.85 | 0.15 |
| 9 | c299 | Middle Eastern and North African countries and cities | 0.99 | 0.79 | 0.2 | 0.99 | 0.78 | 0.21 |
| 9 | c192 | Diversity and Identity | 0.99 | 0.88 | 0.11 | 0.98 | 0.88 | 0.1 |
| 9 | c468 | Russian male names. | 1 | 0.63 | 0.37 | 1 | 0.61 | 0.39 |
| 9 | c588 | Gender-related terms. | 1 | 0.69 | 0.31 | 0.99 | 0.76 | 0.23 |
| 9 | c74 | Financial terms | 0.99 | 0.86 | 0.13 | 0.97 | 0.83 | 0.14 |
| 9 | c503 | Middle East Conflict. | 0.99 | 0.75 | 0.24 | 0.99 | 0.71 | 0.28 |
| 9 | c325 | Violent Crimes | 0.99 | 0.78 | 0.21 | 0.98 | 0.82 | 0.16 |
| 9 | c535 | Academic Research. | 1 | 0.88 | 0.12 | 0.99 | 0.84 | 0.15 |
| 9 | c256 | List of names | 0.98 | 0.76 | 0.22 | 0.98 | 0.74 | 0.24 |
| 9 | c507 | Positive Adjectives | 0.98 | 0.78 | 0.2 | 0.98 | 0.79 | 0.19 |
| 9 | c345 | List of Chinese surnames. | 0.99 | 0.86 | 0.13 | 1 | 0.87 | 0.13 |
| 12 | c259 | List of names | 0.98 | 0.89 | 0.09 | 0.99 | 0.89 | 0.1 |
| 12 | c62 | Adverbs | 0.97 | 0.82 | 0.15 | 0.96 | 0.81 | 0.15 |
| 12 | c128 | Medical and Healthcare Terminology. | 0.99 | 0.8 | 0.19 | 0.98 | 0.82 | 0.18 |
| 12 | c301 | Gender-related nouns and pronouns. | 0.98 | 0.8 | 0.18 | 0.98 | 0.82 | 0.16 |
| 12 | c37 | List of male names. | 0.98 | 0.8 | 0.18 | 0.99 | 0.8 | 0.19 |
| 12 | c281 | Adverbs | 0.99 | 0.8 | 0.19 | 0.99 | 0.78 | 0.21 |
| 12 | c220 | List of surnames. | 0.97 | 0.86 | 0.11 | 0.96 | 0.85 | 0.11 |
| 12 | c432 | List of Male Names | 1 | 0.71 | 0.29 | 0.97 | 0.73 | 0.24 |
| 12 | c439 | Sports commentary terms. | 0.9 | 0.82 | 0.08 | 0.94 | 0.74 | 0.20 |
| 12 | c173 | List of football team names and stadiums. | 0.99 | 0.82 | 0.17 | 0.99 | 0.87 | 0.12 |
| 12 | c348 | List of female names and titles. | 0.99 | 0.75 | 0.24 | 0.98 | 0.7 | 0.28 |
| 12 | c142 | Conflict and War | 0.97 | 0.86 | 0.11 | 0.96 | 0.86 | 0.1 |
| 12 | c245 | Middle East Conflict | 0.99 | 0.76 | 0.23 | 0.98 | 0.72 | 0.26 |
| 12 | c210 | List of male first names | 0.97 | 0.71 | 0.26 | 0.97 | 0.74 | 0.23 |
| 12 | c564 | List of cities and universities in the southeastern United States. | 0.99 | 0.76 | 0.23 | 0.97 | 0.85 | 0.12 |
| 12 | c533 | LGBTQ+ | 1 | 0.71 | 0.29 | 1 | 0.82 | 0.18 |
| 12 | c19 | Complex relationships and interactions between family members and partners. | 0.98 | 0.79 | 0.19 | 0.98 | 0.81 | 0.17 |
| 12 | c263 | Locations in New York City | 0.95 | 0.67 | 0.28 | 0.95 | 0.7 | 0.25 |
| 12 | c487 | List of Spanish male names. | 0.98 | 0.84 | 0.14 | 0.95 | 0.69 | 0.26 |
| 12 | c247 | Scandinavian/Nordic names and places. | 0.98 | 0.77 | 0.21 | 0.97 | 0.75 | 0.22 |
| 12 | c44 | Southeast Asian Politics and Ethnic Conflict | 0.96 | 0.85 | 0.11 | 0.97 | 0.8 | 0.17 |
| 12 | c438 | Verbs for various actions and outcomes. | 0.97 | 0.83 | 0.14 | 0.97 | 0.85 | 0.12 |
| 12 | c421 | Names of people and places in the Middle East | 0.98 | 0.91 | 0.07 | 0.97 | 0.9 | 0.07 |
| 12 | c553 | Islamic Terminology. | 1 | 0.7 | 0.3 | 1 | 0.85 | 0.15 |
| 12 | c149 | List of tennis players' names. | 0.95 | 0.73 | 0.22 | 0.98 | 0.72 | 0.26 |
| 12 | c365 | Criminal activities | 0.95 | 0.77 | 0.18 | 0.97 | 0.82 | 0.15 |

Table 9: Training Probing Classifiers for the concepts shown in Table 8

| Layer | Cluster Tag | Label | val acc | val C acc | sel val | test acc | test c acc | sel test |
|---|---|---|---|---|---|---|---|---|
| 12 | c189 | Superlatives | 0.98 | 0.61 | 0.37 | 0.96 | 0.79 | 0.17 |
| 12 | c248 | Substance abuse. | 0.97 | 0.6 | 0.37 | 1 | 0.81 | 0.19 |
| 12 | c361 | LGBTQ+ and Gender-related Terms | 1 | 0.88 | 0.12 | 1 | 0.9 | 0.1 |
| 3 | c756 | Gender and Sex Labels | 0.87 | 0.72 | 0.15 | 1 | 0.8 | 0.2 |
| 0 | c720 | Gender and Sex Labels | 1 | 0.74 | 0.26 | 1 | 0.55 | 0.45 |
| 0 | c402 | List of female names. | 0.97 | 0.72 | 0.25 | 0.98 | 0.82 | 0.16 |
| 12 | c127 | Geopolitical entities and affiliations. | 0.97 | 0.68 | 0.29 | 0.98 | 0.57 | 0.41 |
| 0 | c707 | Names of US Presidents and Politicians | 1 | 0.55 | 0.45 | 1 | 0.55 | 0.45 |
| 6 | c101 | Speech verbs. | 0.98 | 0.84 | 0.14 | 1 | 0.7 | 0.3 |
| 0 | c820 | Negative adjectives. | 1 | 0.81 | 0.19 | 0.97 | 0.86 | 0.11 |
| 12 | c769 | Food items. | 0.92 | 0.67 | 0.25 | 0.96 | 0.8 | 0.16 |
| 0 | c149 | Fruit and plant-related words. | 1 | 0.7 | 0.3 | 0.95 | 0.81 | 0.14 |
| 3 | c705 | Tourism-related terms | 0.95 | 0.67 | 0.28 | 0.91 | 0.83 | 0.08 |
| 12 | c196 | Verbs of Authority and Request | 0.95 | 0.68 | 0.27 | 0.98 | 0.89 | 0.09 |
| 12 | c398 | Energy sources. | 1 | 0.67 | 0.33 | 1 | 0.69 | 0.31 |
| 6 | c185 | Gender-related terms | 0.98 | 0.64 | 0.34 | 0.96 | 0.68 | 0.28 |
| 3 | c213 | Finance and Taxation. | 0.97 | 0.81 | 0.16 | 0.98 | 0.65 | 0.33 |
| 0 | c92 | Descriptors of geographic regions and types of organizations. | 1 | 0.73 | 0.27 | 0.98 | 0.84 | 0.14 |
| 3 | c659 | Locations in the United States | 1 | 0.88 | 0.12 | 1 | 0.61 | 0.39 |
| 0 | c673 | List of Italian first names. | 1 | 0.93 | 0.07 | 0.89 | 0.8 | 0.09 |
| 3 | c67 | List of male names. | 0.99 | 0.81 | 0.18 | 0.99 | 0.81 | 0.18 |
| 0 | c883 | Nouns | 0.97 | 0.83 | 0.14 | 0.99 | 0.81 | 0.18 |
| 6 | c898 | TV Networks | 1 | 0.68 | 0.32 | 1 | 0.55 | 0.45 |
| 12 | c653 | List of years. | 1 | 0.9 | 0.1 | 1 | 0.91 | 0.09 |
| 0 | c697 | Military Terminology | 1 | 0.62 | 0.38 | 1 | 0.62 | 0.38 |
| 3 | c560 | Political ideologies and systems. | 1 | 0.58 | 0.42 | 0.94 | 0.75 | 0.19 |

Table 10: Probe Results for some concepts chosen from several layers in ALBERT

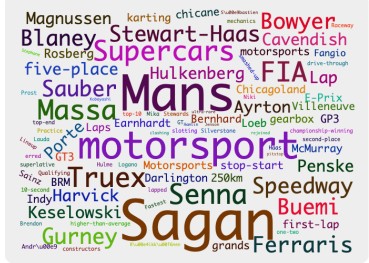

(a) Motorsport Terminology

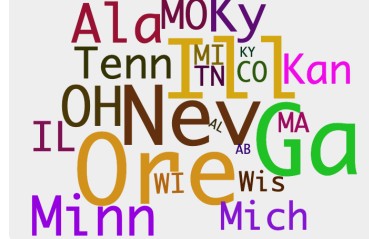

(b) US State Abbreviations

(c) Education-related terms

Figure 10: Example of concepts that were deemed uninterpretable in the BCN but were correctly labeled by ChatGPT

| cluster | label | score |
|---------|-------|-------|
| c55 | Nouns | 0.2 |
| c13 | Nouns | 0.3 |
| c273 | Nouns | 0.1 |
| c268 | Nouns | 0.4 |
| c405 | Nouns | 0.0 |
| c315 | Nouns | 0.3 |
| c231 | Nouns related to various activities and objects | 0.6 |
| c468 | Nouns | 0.2 |
| c524 | Nouns | 0.2 |
| c387 | Nouns | 0.3 |
| c279 | Nouns related to various industries and sectors | 0.6 |
| c440 | Nouns related to various professions and groups | 0.1 |
| c202 | Nouns | 0.3 |
| c237 | Adjectives with no clear category or theme | 0.2 |
| c299 | Adjectives describing attributes of products or services | 0.3 |
| c96 | Adjectives describing ownership, operation or support of various entities and technologies | 0.3 |
| c95 | Adjectives describing various types of related events or phenomena | 0.1 |
| c198 | Adjectives with no clear label | 0.2 |
| c53 | Comparative Adjectives | 0.3 |
| c335 | Comparative Adjectives | 0.2 |
| c531 | Comparative Adjectives | 0.1 |
| c131 | Descriptive/Adjective Labels | 0.4 |
| c505 | Location-based Adjectives | 0.2 |
| c11 | Adjectives describing various types of entities | 0.0 |
| c466 | Adjectives describing ownership, operation, or support of various entities and technologies. | 0.1 |
| c419 | Adjectives describing negative or challenging situations. | 0.6 |
| c128 | Adjectives describing the quality or appropriateness of something. | 0.4 |
| c458 | Adjectives | 0.0 |
| c401 | Comparative Adjectives | 0.1 |
| c444 | Time-related frequency adjectives | 0.0 |
| c52 | Adverbs. | 0.6 |
| c155 | Adverbs of frequency and manner. | 0.3 |
| c136 | Adverbs of degree/intensity. | 0.3 |
| c58 | Adverbs of time and transition. | 0.5 |
| c41 | Adverbs of degree and frequency. | 0.5 |
| c589 | Adverb intensity/degree | 0.2 |
| c265 | Adverbs of Probability and Certainty | 0.3 |
| c251 | Adverbs of frequency and manner. | 0.2 |
| c57 | Adverbs of Frequency | 0.4 |
| c555 | Temporal Adverbs. | 0.4 |
| c302 | Frequency Adverbs | 0.2 |
| c332 | Adverbs of manner and opinion. | 0.4 |
| c546 | Adverbs of degree/intensity. | 0.3 |
| c570 | Adverbs of preference/choice. | 0.5 |
| c244 | Adverbs indicating degree or extent. | 0.3 |
| c222 | Adverbs of Time | 0.5 |
| c309 | Adverbs describing degree or intensity. | 0.2 |
| c487 | List of numerical values. | 0.2 |
| c179 | Numerical Data. | 0.3 |
| c420 | Numerical data. | 0.2 |
| c390 | List of numbers | 0.3 |
| c287 | Numeric Data. | 0.0 |
| c101 | List of numerical values. | 0.5 |
| c494 | List of numerical values. | 0.5 |
| c579 | Numerical data. | 0.2 |
| c537 | List of numerical values. | 0.2 |
| c435 | Numerical data. | 0.3 |
| c528 | List of numerical values. | 0.3 |
| c549 | List of prices. | 0.5 |
| c398 | Numerical Data. | 0.0 |
| c359 | List of numerical values. | 0.1 |
| c477 | List of monetary values. | 0.1 |
| c593 | List of monetary values. | 0.1 |
| c80 | Numeric quantities. | 0.1 |

Table 11: Neuron Analysis Results on *Super Concepts* extracted from BERT-base-cased model. The alignment column shows the intersection between the top 10 neurons in the Super concept and the Sub concepts.

| cluster | label | score |
| --- | --- | --- |
| c259 | List of names | 0.4 |
| c37 | List of male names. | 0.3 |
| c328 | List of names of politicians, public figures, and athletes. | 0.2 |
| c220 | List of surnames. | 0.4 |
| c433 | List of names. | 0.5 |
| c262 | List of surnames | 0.4 |
| c210 | List of male first names. | 0.1 |
| c231 | List of female names. | 0.4 |
| c383 | List of names | 0.2 |
| c280 | List of names. | 0.2 |
| c202 | List of surnames. | 0.2 |
| c344 | Irish surnames | 0.3 |
| c6 | Surnames | 0.4 |
| c75 | List of female names. | 0.7 |
| c269 | List of celebrity names | 0.4 |
| c578 | List of surnames. | 0.2 |
| c535 | List of names | 0.2 |
| c487 | List of Spanish male names. | 0.2 |
| c340 | Last names. | 0.4 |
| c48 | List of surnames. | 0.0 |
| c70 | List of names. | 0.1 |
| c353 | List of names in the entertainment industry. | 0.2 |
| c568 | List of names. | 0.4 |
| c378 | List of surnames. | 0.1 |
| c575 | Surnames | 0.4 |
| c149 | List of tennis players' names. | 0.4 |
| c325 | List of names. | 0.2 |
| c436 | List of sports players' names | 0.2 |
| c594 | List of surnames. | 0.6 |

Table 12: Name clusters extracted from the last layer of BERT-base-cased