# OpenReview forum: "Can LLMs Facilitate Interpretation of Pre-trained Language Models?"
_EMNLP/2023/Conference — EMNLP 2023 Main_

### Official Review · Reviewer_ZW2Y · 2023-08-04

**Typos Grammar Style And Presentation Improvements:** 1. The clarity between lines 146-153 …
**Soundness:** 4

**Excitement:**

4: Strong: This paper deepens the understanding of some phenomenon or lowers the barriers to an existing research direction.

**Paper Topic And Main Contributions:**

This paper addresses the challenge of interpreting the latent concepts learned by pre-trained language models (pLMs). The authors specifically focus on the task of latent concept learning, which involves labeling high-level concepts to describe clusters of words derived from pre-trained language models. They input sentences into pre-trained language models and cluster the embeddings from each layer to identify latent concepts.

Large language models, specifically ChatGPT, are utilized to generate concept labels, which are then validated for their acceptability and precision. The primary contribution of this paper is the scaling up of concept labels, which facilitates the analysis of concepts across multiple pre-trained language models and the exploration of the relationship between neuron activation and fine-grained linguistic phenomena. This paper contributes to the field of computationally-aided linguistic analysis by demonstrating the potential of large-language models, such as ChatGPT, in interpreting smaller pre-trained language models.

**Questions For The Authors:**

A: Have you investigated the differences across layers? Did you observe any variations in concepts from different levels?

B: How does the concept clustering process, including the choice of clustering algorithms and hyperparameters, impact the overall probing process?

C: Could you provide more clarity on the training process of the binary classifier? The section on Concept Probing (sec 2.3) is challenging to comprehend, particularly regarding the relationships between word clusters, annotated concepts, sentences containing the word clusters, and how they are utilized to extract features. It would be helpful to clarify how contextualized feature vectors are obtained.

**Reasons To Accept:**

The interpretability of language models is a crucial research area, and this paper offers a unique approach by utilizing large-language models as annotators to enhance the interpretability of pre-trained language models.

- The use of large-language models as annotators enables the generation of precise labels for concepts within pre-trained language models. While this idea is not entirely new, the validation and analysis presented in this paper are intriguing.
- Additionally, the release of the resource Transformers Concept-Net, containing 39K annotated concepts from five pre-trained language models, will facilitate future research in the field of interpretability.
- This approach can potentially be extended to probe concepts in other open-source large language models, thus aiding the community in understanding the differences across various language models.

**Reasons To Reject:**

I wouldn't recommend rejecting the paper if the author addresses all the raised questions and improves the paper's readability. However, there are a few potential minor weaknesses to consider:

- The comparison of ChatGPT annotations with human-annotated concepts is based on a small sample size of 269 concepts, which might not encompass all possible scenarios or complexities. One significant aspect of latent concepts, their hierarchy or levels of abstractness, isn't discussed in this paper.
- The findings from this paper are based on the use of ChatGPT as an annotator. The degree to which these findings can be generalized to other language models remains uncertain. However, given the focus on ChatGPT, this could be an area for future exploration.

**Reproducibility:**

4: Could mostly reproduce the results, but there may be some variation because of sample variance or minor variations in their interpretation of the protocol or method.

**Reviewer Confidence:**

4: Quite sure. I tried to check the important points carefully. It's unlikely, though conceivable, that I missed something that should affect my ratings.

---

> ### Author Rebuttal · Authors · 2023-08-29
>
> **R: The comparison of ChatGPT annotations with human-annotated concepts is based on a small sample size of 269 concepts, which might not encompass all possible scenarios or complexities.**
>
> A: We agree that our evaluation of 269 concepts may not cover all scenarios, but our aim was to compare LLM annotations to gold standard human annotations. Creating a human-annotated concept database like BCN is costly and time-consuming. Our study shows LLM annotations as a viable, cost-effective, and scalable alternative.
>
> **R: One significant aspect of latent concepts, their hierarchy or levels of abstractness, isn't discussed in this paper.**
>
> A: We touched upon this briefly while presenting our results in the Neuron Analysis (Section 5.2).  We showcased how our LLM annotation unveils the underlying sub-concepts of a super-concept within the latent space. Consequently, this enables a comprehensive examination of neurons, providing more refined insights into the foundational constituents of these concepts. Exploring this avenue further could yield intriguing possibilities, perhaps even modifying the prompt to the LLM to encode each concept in broader and finer categories leading to a form of hierarchy.
>
> **R: The findings from this paper are based on the use of ChatGPT as an annotator. The degree to which these findings can be generalized to other language models remains uncertain. However, given the focus on ChatGPT, this could be an area for future exploration.**
>
> A: While the extent to which these findings can be extended to other language models presents uncertainty, the trajectory of advancements in LLMs suggests that future models will likely be even more potent. We remain hopeful that these improvements will enable us to attain even more refined annotations of latent concepts. What would be interesting to see is whether less powerful models can achieve comparable effectiveness in annotating latent spaces, especially if our aim is to reduce inference costs. This avenue of exploration holds promising potential for further research.
>
> **R: Question A: Have you investigated the differences across layers? Did you observe any variations in concepts from different levels?**
>
> A: In our layer-wise analysis of concepts, we discovered that the concepts in initial layers predominantly focussed on word morphology and reconstructing word structure that is disintegrated due to sub-word segmentation. Consequently many concepts composed of words grouped based on common ngrams are found. But deeper within the layers, we found a mixture of concepts based on morphological, semantic and sometimes syntactic similarities. In the final layers we note that the concepts became increasingly aligned with the downstream tasks for which the model was trained. For instance, we began to observe concepts encompassing positively and negatively polarized words, indicative of sentiment classification tasks. Similarly, clusters formed in accordance with POS tags within models tailored for predicting such linguistic features. We excluded this analysis from the paper due to space limitations and its limited coherence with the overall content.
>
> **R:  Question B: How does the concept clustering process, including the choice of clustering algorithms and hyperparameters, impact the overall probing process?**
>
> A: We have explored agglomerative clustering with varying cluster sizes, ranging from 600 to 1000. Additionally, we've been investigating alternative clustering algorithms like K-means, aiming to replace the resource-intensive agglomerative clustering approach. In our results, we have found the resulting concepts from different parameters and algorithms to be similarly aligned with human-defined concepts, and have not seen these to have any impact on the overall probing process.
>
> **R:  Question C: Could you provide more clarity on the training process of the binary classifier? The section on Concept Probing (sec 2.3) is challenging to comprehend, particularly regarding the relationships between word clusters, annotated concepts, sentences containing the word clusters, and how they are utilized to extract features. It would be helpful to clarify how contextualized feature vectors are obtained.**
>
> A: After identifying an encoded concept through the process of Concept Discovery (Section 2.1) and assigning it a label C via Concept Annotation (Section 2.2), we assemble data points in the format <z_i^l, C>, where z_i^l signifies a contextualized feature of word w_i ∈ C in layer l. This feature captures the embedding of the word w_i within a sentence, derived by conducting a forward pass through the network up to layer l. To construct a training dataset for the binary classifier, we replicate the positive samples data size for the negative class. This is accomplished by extracting feature vectors using words that are not part of the concept C, denoted as ∉C. By combining these two sets of data, we amass the necessary information to effectively train a binary probe. Both probing classifiers and neuron analysis is carried out using the same data. We will rewrite and expand on details in the paper.
>
> **R: The clarity between lines 146-153 needs further improvement. Could you provide more information on the input and output? Are the representation for 5M tokens from 12 layers being clustered?**
>
> A: We run clustering for each layer independently. This enables us to analyze the types of concepts learned within various layers of the network.
>
> **R:  The use of "ConceptNet" in the abstract (line 022) is confusing, considering the widespread use of the knowledge graph ConceptNet.**
>
> A: We intend to use _Transformers Concept Net_ for our repository of concepts, indicating that this is a collection of concepts specifically associated with a class of models (i.e. transformers).

---

### Official Review · Reviewer_zd3p · 2023-08-11

**Soundness:** 4

**Excitement:**

3: Ambivalent: It has merits (e.g., it reports state-of-the-art results, the idea is nice), but there are key weaknesses (e.g., it describes incremental work), and it can significantly benefit from another round of revision. However, I won't object to accepting it if my co-reviewers champion it.

**Paper Topic And Main Contributions:**

This paper explored to what extent LLMs capture linguistic concepts. They focus on two unexplored commonsense domains -- taste and physical properties. The authors used ChatGPT to scale up the annotation process of semantic concepts and show that it is comparable to human quality. They compare different LLMs in terms of concept understanding in the two mentioned domains and find that BERT is surprisingly good compared to the much larger GPT-based models. They will also release their dataset.

**Questions For The Authors:**

* Your definition of feature vectors is somewhat unclear. Which layer does the upper (l) notation for z denote?
* What parameters were used for the agglomerative clustering?
* Why do you use a number of clusters as stopping criteria instead of a similarity threshold, which should capture better the dynamic nature of human linguistic concepts?
* Line 204: What is S_i? Did you mean D_i?
* Line 138: What do you mean by "word type"?
* Table 1: 0.34 is not a good agreement (even for hard tasks), it's fair at most. Personally, I wouldn't trust annotations with this score and continue to formalize the subjective aspects of the annotation task (as much as possible, I don't expect perfect agreement).
* The neuron analysis is hard to understand. What are the scores in Table 4?



**Reasons To Accept:**

* This is an interesting approach to shifting the annotation load from humans to LLMs and it can be useful in other domains as well.
* Our current understanding of the knowledge encoded by LLMs is lacking. Thus, this is a relevant research direction.
* The usage of the classifier on top of the feature vectors is a nice idea.

**Reasons To Reject:**

* It was hard for me to understand the neuron analysis. Rewriting it in a form that will allow readers who are not familiar with that type of analysis to understand is required. Moreover, I cannot trust my review regarding this part, as it hard to criticize something I do not fully understand. If the authors would clear this part in the rebuttal I could modify my review accordingly.
*  The contribution of the analysis, while interesting, is rather limited (two very specific domains).

**Reproducibility:**

3: Could reproduce the results with some difficulty. The settings of parameters are underspecified or subjectively determined; the training/evaluation data are not widely available.

**Reviewer Confidence:**

2: Willing to defend my evaluation, but it is fairly likely that I missed some details, didn't understand some central points, or can't be sure about the novelty of the work.

**Typos Grammar Style And Presentation Improvements:**

* I'd probably make a section for each type of analysis. Switching from concept vectors to neurons is a bit confusing. Having one section for each flows better.
* The explanations of both "Probing Framework" and "Neuron Analysis" are not very clear to a reader who is unfamiliar with these terms.
* Line 140: use the pLMs acronym.
* The paragraph referring to the results in Table 1 does not states that the precision is 67.6%. For the sake of completeness please write it.
* Please note you use the acronym BCN before defining it.

---

> ### Author Rebuttal · Authors · 2023-08-29
>
> **R: It was hard for me to understand the neuron analysis. Rewriting it in a form that will allow readers who are not familiar with that type of analysis to understand is required. Moreover, I cannot trust my review regarding this part, as it is hard to criticize something I do not fully understand. If the authors would clear this part in the rebuttal I could modify my review accordingly.**
>
> A: We had to trim down the details due to space limitations. Hopefully, with the additional page provided for the final version, we will be able to provide a more comprehensive explanation to enhance readers' understanding. Below, we attempt to further explain this section of the paper:
> In contrast to the holistic view offered by representation analysis, neuron analysis highlights the **role of individual neurons** (or groups of them) within a neural network. The methodologies employed in neuron analysis (please refer to Sajjad et al., 2022) typically require human-in-the-loop intervention to comprehend a neuron's function (e.g., Karpathy et al., 2015, Kadar et al., 2017, etc.), making their analysis infeasible to scale. Alternatively, they rely on linguistic annotations of predefined concepts (e.g., Lakretz et al., 2019, Hennigen et al., 2020, Mu and Andreas, 2020). Consequently, the resulting neuron explanations are subject to the same limitations we address in this study.
>
> Our research demonstrates that annotating the latent space using LLM (Large Language Model) overcomes these constraints. We conducted neuron analysis on the latent space of the BERT model using the Probeless method (Antverg and Belinkov, 2022). This analysis unveiled neurons that capture intricate semantic hierarchies — such as those distinguishing adverbs of time from adverbs of frequency, those encoding price-related numbers, and those capturing monetary value indicators (Please refer to Figure 4). This method can be seamlessly applied to any encoded concept by leveraging its GPT label, facilitating the discovery of neurons associated with that concept. In summary, our approach obviates the need for human intervention or annotated data to make such discoveries, addressing a limitation that hampers other interpretation methods.
>
> **R: The contribution of the analysis, while interesting, is rather limited (two very specific domains).**
>
> A: While our demonstration applies to enhancing Probing and Neuron Analysis interpretation frameworks, we firmly believe that our findings hold potential to advance the broader landscape of NLP model explainability, for instance:
>
> - It is possible to extend the TCAV framework (Kim et al., 2018) beyond the confines of predefined concepts to the entire latent space of the model.
> - Corpus-based methods explain neurons by leveraging the concept of shared ngrams, often using linguistic concepts (Na et al. 2019) or manual annotations (Kadar et al. 2017). Our approach can be seamlessly applied to these existing methodologies.
> - The ongoing research in latent ontology induction (Michael et al., 2020, and Fu and Lapata, 2022) can take advantage from the incorporation of LLM annotation into their methodology.
>
>
> **R: Your definition of feature vectors is somewhat unclear. Which layer does the upper (l) notation for z denote?**
>
> A: It doesn't refer to a particular layer, but rather to any layer "l" within the network. Each layer encodes a distinct array of concepts.
>
> **R: What parameters were used for the agglomerative clustering?
> R: Why do you use a number of clusters as stopping criteria instead of a similarity threshold, which should capture better the dynamic nature of human linguistic concepts?**
>
> A: Our aim in this work was to evaluate the efficacy of LLMs as annotators of latent spaces, and hence we relied on the settings prescribed  in Dalvi et al., (2022). We used 600 clusters applying Ward’s minimum variance criterion that minimizes the total within-cluster variance. In their exploration Dalvi et al., found alternative methods such as ELbow and Silhouette to not yield reliable results.
>
> **R: Line 204: What is S_i? Did you mean D_i?**
>
> A: S_i denotes the ith sentence within dataset D. Due to limitations on sentence length during inference, we perform a forward pass over individual sentences to acquire the corresponding contextual feature vectors
>
> **R: Line 238: What do you mean by "word type"?**
>
> A: We are referring to a unique variation of a word, essentially an element of the vocabulary of the dataset. We consider ten distinct contexts per word type.
>
> **R: Table 1: 0.34 is not a good agreement (even for hard tasks), it's fair at most. Personally, I wouldn't trust annotations with this score and continue to formalize the subjective aspects of the annotation task (as much as possible, I don't expect perfect agreement).**
>
> A: We apologize for the typographical error on line 305. Our intention was to denote 'Fair' (as also evident in Table 1) and not 'Fair to good.' We will rectify the error. Additionally, we intend to elaborate on the complexity and subjectivity associated with annotating Q2. This complexity varies depending on the annotator's knowledge and their subjective perspective regarding the boundary between precise and imprecise labels.
>
> **R: The neuron analysis is hard to understand. What are the scores in Table 4?**
>
> A: We have tried to explain the neuron analysis section in the first comment. Did you mean Table 4 (Probing Analysis Results) or Table 5 (Neuron Analysis)?
>
> Table 4 is showing accuracy numbers for probing classifiers (representation analysis) when they are trained towards predicting different encoded concepts. Classifier accuracy is used as a measure to demonstrate how strongly a concept is represented within the latent space of a model. We use this to carry out a comparative analysis among ALBERT and XLNet models in Table 4 and with 3 other models (BERT, XLM-R and RoBERTa) in Table 7 (Appendix B).
>
> Table 5 along with its corresponding Figure 4 present the outcomes of our neuron analysis. In this analysis, we showcase how neurons within a super concept (such as all adverbs in POS tagging) are distributed among sub-concepts (various forms of adverbs). This distribution is quantified by the percentage of overlap between neurons from the sub-concepts with those from the super-concept. Our LLM annotation reveals the sub-concepts inherent to a super-concept within the latent space. This, in turn, facilitates an in-depth neuron analysis, offering more precise insights into the underlying neurons that constitute these concepts.
>
> Thank you for your feedback regarding the typos and unclear sections in the paper. We are committed to addressing these concerns and improving clarity in the camera-ready version.
>
> ### References:
>
> Hassan Sajjad, Nadir Durrani, and Fahim Dalvi. 2022. Neuron-level Interpretation of Deep NLP Models: A Survey. Transactions of the Association for Computational Linguistics.
>
> Andrej Karpathy, Justin Johnson, and Li Fei-Fei. 2015. Visualizing and understanding recurrent networks. arXiv preprint arXiv:1506.02078.
> Akos K´ad´ar, Grzegorz Chrupała, and Afra ´Alishahi. 2017. Representation of linguistic form and function in recurrent neural networks. Computational Linguistics, 43(4):761–780.
>
> Yair Lakretz, German Kruszewski, Theo Desbordes, Dieuwke Hupkes, Stanislas Dehaene, and Marco Baroni. 2019. The emergence of number and syntax units in LSTM language models. In Proceedings of the 2019 Conference of the North American Chapter of the Association for Computational Linguistics: Human Language Technologies, Volume 1 (Long and Short Papers), pages 11–20, Minneapolis, Minnesota. Association for Computational Linguistics.
>
> Lucas Torroba Hennigen, Adina Williams, and Ryan Cotterell. 2020. Intrinsic probing through dimension selection. In Proceedings of the 2020 Conference on Empirical Methods in Natural Language Processing (EMNLP), pages 197–216, Online. Association for Computational Linguistics.
>
> Jesse Mu and Jacob Andreas. 2020. Compositional explanations of neurons. NeurIPS 2020
> Omer Antverg and Yonatan Belinkov. 2021. On the pitfalls of analyzing individual neurons in language models. In International Conference on Learning Representations.
>
> Kim, B.; Wattenberg, M.; Gilmer, J.; Cai, C.; Wexler, J.; Viegas, F.; et al. 2018. Interpretability beyond feature attribution: Quantitative testing with concept activation vectors (TCAV). In Proceedings of the International Conference on Machine Learning, 2668–2677. JMLR.
>
> Seil Na, Yo Joong Choe, Dong-Hyun Lee, and Gunhee Kim. 2019. Discovery of natural language concepts in individual units of CNNs, ICLR 2019
>
> Julian Michael, Jan A. Botha, and Ian Tenney. 2020. Asking without telling: Exploring latent ontologies
> in contextual representations. In Proceedings of the 2020 Conference on Empirical Methods in Natural Language Processing, EMNLP ’20, pages 6792–6812, Online. Association for Computational Linguistics.
>
> Yao Fu and Mirella Lapata. 2022. Latent topology induction for understanding contextualized representations.

---

### Official Review · Reviewer_VLTK · 2023-08-12

**Typos Grammar Style And Presentation Improvements:** No
**Soundness:** 4

**Excitement:**

3: Ambivalent: It has merits (e.g., it reports state-of-the-art results, the idea is nice), but there are key weaknesses (e.g., it describes incremental work), and it can significantly benefit from another round of revision. However, I won't object to accepting it if my co-reviewers champion it.

**Missing References:**

No

**Paper Topic And Main Contributions:**

**Topics**
* Interpretation and analysis of pre-trained language models (LMs)
* Using large language models (LLMs) like ChatGPT as annotators for latent concepts
* Enabling fine-grained probing and neuron analysis with LLM annotations

**Main Contributions**
* Show that ChatGPT provides accurate and semantically rich annotations for latent concepts learned by LMs
* Demonstrate that GPT annotations empower fine-grained probing for concepts beyond human-defined categories
* Illustrate how GPT annotations enhance neuron analysis towards more intricate concepts
* Contribute a framework for using LLMs to facilitate interpretation and analysis of LMs

In sum, the paper explores using ChatGPT to annotate latent concepts discovered through clustering of LM representations. It then leverages these annotations to enable more detailed probing and neuron analysis. The main contribution is illustrating how LLMs can be utilized to interpret and analyze pre-trained LMs at a finer granularity than enabled by human annotations.


**Questions For The Authors:**

1. It is mentioned that previous interpretation analysis techniques rely on having access to annotated corpora or human involvement. Could you elaborate more on the limitations this imposes? For example, does it restrict the scope or scale of analysis that can be performed?

2. In the error analysis, several factors are identified affecting ChatGPT's performance like sensitive content, lexical properties, insufficient context etc. Do you have any ideas on how to further improve the prompts or methodology to address these?

3. One concern could be the computational expense of querying ChatGPT extensively. Do you have any ideas to make this more efficient or cost-effective at scale?


**Reasons To Accept:**

* The paper tackles an important problem in NLP research - interpreting and analyzing pre-trained language models. Understanding what knowledge these powerful models encode is crucial.
* The key idea of using LLMs like ChatGPT as annotators to label latent concepts is novel and has not been explored before. This enables scaling up interpretation analysis.
* Thorough experiments compare ChatGPT annotations to human annotations, demonstrating its accuracy and richer semantic labeling.
* The paper clearly shows the benefits of using ChatGPT annotations through two applications - probing and neuron analysis. This provides a proof of concept.
* Releasing a large dataset of 39K annotated concepts will facilitate further research in this exciting direction.
* The methodology is sound, combining concept discovery through clustering with prompt engineering for annotations. Evaluation is rigorous.
* The writing is clear and easy to follow. The introduction provides sufficient background. Experiments and results are well explained.
* Limitations regarding computational expense, content filtering, and concept drift in LLMs are properly acknowledged.


**Reasons To Reject:**

* The main contribution and novelty are not clearly articulated. The authors claim they are using ChatGPT to annotate latent concepts in pretrained language models, but do not sufficiently justify why this is an important advancement compared to prior work using human annotations or predefined ontologies. The potential benefits of ChatGPT annotations are hinted at but not convincingly showcased.

* The methodology lacks key details. The concept discovery process using clustering is described briefly but specifics like the dataset, clustering algorithm parameters, number of clusters etc are missing. The process of prompting ChatGPT for annotations is also unclear - what was the exact prompt, how were hyperparameters like temperature set, how many examples were annotated.


**Reproducibility:**

3: Could reproduce the results with some difficulty. The settings of parameters are underspecified or subjectively determined; the training/evaluation data are not widely available.

**Reviewer Confidence:**

4: Quite sure. I tried to check the important points carefully. It's unlikely, though conceivable, that I missed something that should affect my ratings.

---

> ### Author Rebuttal · Authors · 2023-08-29
>
> **R: The main contribution and novelty are not clearly articulated. The authors claim they are using ChatGPT to annotate latent concepts in pretrained language models, but do not sufficiently justify why this is an important advancement compared to prior work using human annotations or predefined ontologies. The potential benefits of ChatGPT annotations are hinted at but not convincingly showcased.
> R:  It is mentioned that previous interpretation analysis techniques rely on having access to annotated corpora or human involvement. Could you elaborate more on the limitations this imposes? For example, does it restrict the scope or scale of analysis that can be performed?**
>
> A: The use of predefined linguistic concepts and human-in-the-loop methods are restrictive in different ways:
>
> - Pre-defined ontologies result in a narrow view of the model's knowledge.
> - Human-in-the-loop are both costly and subjective impeding scalability
>
> Dalvi et al., (2022) showed that the pre-trained language models learn novel concepts which _do not strictly adhere to any predefined linguistic groups_. However, manual annotations of these concepts are costly, making the human-in-the-loop infeasible. For an accurate understanding of decision making in these models, we must understand how knowledge is preserved within the latent space using large scale annotations of the encoded concepts. Our research showcases that using an LLM overcomes this hurdle, offering comprehensive and highly accurate linguistic annotations for latent concepts acquired within deep NLP models. While our demonstration applies to enhancing Probing and Neuron Analysis interpretation frameworks, we firmly believe that our findings hold potential to advance the broader landscape of NLP model explainability, for instance Kim et al, 2018, Geva et al., 2021.
>
> **R: The methodology lacks key details. The concept discovery process using clustering is described briefly but specifics like the dataset, clustering algorithm parameters, number of clusters etc are missing. The process of prompting ChatGPT for annotations is also unclear - what was the exact prompt, how were hyperparameters like temperature set, how many examples were annotated.**
>
> A:  We had to condense the details due to space limitations. Nevertheless, specific details can be found in the paper:
>
> **Data details** can be found in **Section 3: Latent Concept Data (First paragraph: l234).**
>
> **Details about the clustering** algorithm parameters are elaborated upon in **Section 3: Concept Discovery (Second paragraph: l246)**. Specifically, we employed an agglomerative hierarchical clustering approach with 600 clusters. We apply Ward’s minimum variance criterion that minimizes the total within-cluster variance.
>
> **The exact prompt used:** please refer to **Section 2.2 Concept Annotation (l180).** The exact prompt is provided at the conclusion of this section, while additional prompts tested during the study are listed in Appendix A.1.
>
> **ChatGPT hyperparameters: Section 3 Concept Annotation (Third paragraph: l251)** contains the relevant information. We used a temperature of 0 to control output randomness and generate deterministic responses. We used the default value (0.95) for the top p.
>
> **The number of annotated examples:** We employed GPT to annotate 39K concepts across 5 pre-trained language models **(Section 5.1, last paragraph: l418).** Additionally, to perform a qualitative comparison of ChatGPT's annotations to human annotations of latent concepts, we annotated 269 concepts within the final layer of BERT-base-cased. We will mention this more clearly in the paper.
>
> **R: In the error analysis, several factors are identified affecting ChatGPT's performance like sensitive content, lexical properties, insufficient context etc. Do you have any ideas on how to further improve the prompts or methodology to address these?**
>
> A: Sensitive Content: OpenAI has introduced a [moderation endpoint tool](https://platform.openai.com/docs/guides/moderation) that, based on the input, identifies which content policy the input violates. We can use the classification label from the tool as a coarse label for the encoded concept. Alternatively, we could explore a version of LLM where content policy models are deactivated.
>
> Lexical Properties: To tackle this issue, we propose creating multiple diverse prompts that capture specific linguistic properties, in addition to the generic prompt. This approach enables us to offer multiple explanations for a latent concept. In Appendix A2-A3, we demonstrated how prompts can be adjusted to encompass specific properties like common ngrams or prevalent parts of speech tags between words.
>
> Providing Context: Our findings indicate that utilizing contextual information (up to 10 sentences) with LLM enhances annotation accuracy. Refer to Figure 3c for an example. For larger concepts with numerous words, a cost-effective strategy could involve offering neighboring 5-grams for each word rather than complete sentences, or subsampling the words for which context is provided.
>
> **R: One concern could be the computational expense of querying ChatGPT extensively. Do you have any ideas to make this more efficient or cost-effective at scale?**
>
> A: An idea might involve limiting the number of words in an underlying concept for making queries. We haven't yet investigated the alterations in annotation accuracy while using a smaller concept subset for querying. Another complementary approach could involve implementing caching and maintaining a concept database along with labels. If a significant overlap is found between the concept to be queried and concepts already existing in the database, it may be feasible to leverage the existing concept's label or opt for a significantly smaller subset of the concept during the querying process.
>
> ### References:
>
> Kim, B.; Wattenberg, M.; Gilmer, J.; Cai, C.; Wexler, J.; Viegas, F.; et al. 2018. Interpretability beyond feature attribution: Quantitative testing with concept activation vectors (TCAV). In Proceedings of the International Conference on Machine Learning, 2668–2677. JMLR.
>
> Mor Geva, Roei Schuster, Jonathan Berant, and Omer Levy. 2021. Transformer feed-forward layers are key-value memories. In Proceedings of the 2021 Conference on Empirical Methods in Natural Language Processing, pages 5484–5495, Online and Punta Cana, Dominican Republic. Association for Computational Linguistics

---

### Meta-Review · Area_Chair_mFiz · 2023-09-13

**Recommendation:** 5

**Metareview:**

Quality, clarity, originality and significance

TLDR: The paper proposes using LLMs, in particular, ChatGPT, to discover/label latent concepts in pre-trained language models. The LLM helps to find semantically richer annotations compared to human annotations. The resulting resource with annotated concepts is successfully used to perform interpretation analysis with both probing and neuron interpretation. The reviewers appreciate the importance and novelty of the studied problem and find the resulting resources and analysis framework facilitating future work in the area. The cons are about improving the clarity of the writing and the limitations of their work, which the authors have already readily acknowledged.


Pros:
1. Significance 1 - all reviewers agree that the paper addresses an important problem - interpreting pre-trained language models.
2. Originality - all reviewers appreciate the novelty of the contributions made in the paper -- using LLMs as annotators to label latent concepts and find the work can be used in other domains/pre-trained models/LLMs as well.
3. Significance 2 - the released resource of annotated concepts will facilitate further research in this direction (VLTK, ZW2Y).
4. Quality - VLTK finds the methodology to be  sound, combined with a rigorous evaluation.
5. Clarity - VLTK also finds the writing to be clear and easy to follow.

Cons:
1. Clarity - all reviewers find the neuron analysis hard to understand, but the authors already give more details in the rebuttal and indicate they'll include them in the final version as well.
2. Clarity/quality - some reviewers identified certain limitations in the analysis performed in the paper, which could be addressed in the limitations section as well, and the authors acknowledge these already in their rebuttal -- using two very specific domains (zd3p), comparison of annotations with human-annotated concepts is based on a small sample size set (ZW2Y), the findings are based only on the use of ChatGPT as an annotator (ZW2Y).

---

### Decision · Program_Chairs · 2023-10-07

**Decision:**

Accept-Main

**Comment:**

Quality, clarity, originality and significance

TLDR: The paper proposes using LLMs, in particular, ChatGPT, to discover/label latent concepts in pre-trained language models. The LLM helps to find semantically richer annotations compared to human annotations. The resulting resource with annotated concepts is successfully used to perform interpretation analysis with both probing and neuron interpretation. The reviewers appreciate the importance and novelty of the studied problem and find the resulting resources and analysis framework facilitating future work in the area. The cons are about improving the clarity of the writing and the limitations of their work, which the authors have already readily acknowledged.


Pros:
1. Significance 1 - all reviewers agree that the paper addresses an important problem - interpreting pre-trained language models.
2. Originality - all reviewers appreciate the novelty of the contributions made in the paper -- using LLMs as annotators to label latent concepts and find the work can be used in other domains/pre-trained models/LLMs as well.
3. Significance 2 - the released resource of annotated concepts will facilitate further research in this direction (VLTK, ZW2Y).
4. Quality - VLTK finds the methodology to be  sound, combined with a rigorous evaluation.
5. Clarity - VLTK also finds the writing to be clear and easy to follow.

Cons:
1. Clarity - all reviewers find the neuron analysis hard to understand, but the authors already give more details in the rebuttal and indicate they'll include them in the final version as well.
2. Clarity/quality - some reviewers identified certain limitations in the analysis performed in the paper, which could be addressed in the limitations section as well, and the authors acknowledge these already in their rebuttal -- using two very specific domains (zd3p), comparison of annotations with human-annotated concepts is based on a small sample size set (ZW2Y), the findings are based only on the use of ChatGPT as an annotator (ZW2Y).